# Single-Loop Stochastic Algorithms for Difference of Max-Structured Weakly Convex Functions

**Quanqi Hu** [1]    **Qi Qi** [2]    **Zhaosong Lu** [3]    **Tianbao Yang** [1]

[1] Department of Computer Science & Engineering, Texas A&M University
[2] Department of Computer Science, The University of Iowa
[3] Department of Industrial and Systems Engineering, University of Minnesota
{quanqi-hu, tianbao-yang}@tamu.edu   qi-qi@uiowa.edu   zhaosong@umn.edu

## Abstract

In this paper, we study a class of non-smooth non-convex problems in the form of $\min_x[\max_{y\in\mathcal{Y}}\phi(x,y) - \max_{z\in\mathcal{Z}}\psi(x,z)]$, where both $\Phi(x) = \max_{y\in\mathcal{Y}}\phi(x,y)$ and $\Psi(x) = \max_{z\in\mathcal{Z}}\psi(x,z)$ are weakly convex functions, and $\phi(x,y), \psi(x,z)$ are strongly concave functions in terms of $y$ and $z$, respectively. It covers two families of problems that have been studied but are missing single-loop stochastic algorithms, i.e., difference of weakly convex functions and weakly convex strongly-concave min-max problems. We propose a stochastic Moreau envelope approximate gradient method dubbed SMAG, the first single-loop algorithm for solving these problems, and provide a state-of-the-art non-asymptotic convergence rate. The key idea of the design is to compute an approximate gradient of the Moreau envelopes of $\Phi, \Psi$ using only one step of stochastic gradient update of the primal and dual variables. Empirically, we conduct experiments on positive-unlabeled (PU) learning and partial area under ROC curve (pAUC) optimization with an adversarial fairness regularizer to validate the effectiveness of our proposed algorithms.

## 1   Introduction

In this paper, we consider a class of non-convex, non-smooth problems in the following form

$$\min_{x\in\mathbb{R}^{d_x}} \left\{ F(x) := \max_{y\in\mathcal{Y}}\phi(x,y) - \max_{z\in\mathcal{Z}}\psi(x,z) \right\}, \tag{1}$$

where the sets $\mathcal{Y} \subset \mathbb{R}^{d_y}$, $\mathcal{Z} \subset \mathbb{R}^{d_z}$ are convex and compact, and the two component functions $\phi(x,y)$ and $\psi(x,z)$ are weakly-convex in terms of $x$ and strongly-concave in the terms of $y$ and $z$, respectively. Both component functions are in expectation forms, i.e., $\phi(x,y) = \mathbb{E}_{\xi\sim\mathcal{D}_\phi}[\phi(x,y;\xi)]$ and $\psi(x,z) = \mathbb{E}_{\zeta\sim\mathcal{D}_\psi}[\psi(x,z;\zeta)]$. We refer to this class of problems as the Difference of Max-Structured Weakly Convex Functions (DMax) Optimization. DMax optimization unifies two emerging families of problems in optimization field, difference-of-weakly-convex (DWC) optimization

$$\min_{x\in\mathbb{R}^{d_x}} \left\{ F(x) := \phi(x) - \psi(x) \right\}, \tag{2}$$

and weakly-convex-strongly-concave (WCSC) min-max optimization

$$\min_{x\in\mathbb{R}^{d_x}} \left\{ F(x) := \max_{y\in\mathcal{Y}}\phi(x,y) \right\}. \tag{3}$$

Thus, DMax optimization has a wide range of applications in machine learning and AI, including applications of DWC optimization (e.g., positive-unlabeled (PU) Learning [39], non-convex

---

Correspondence to: Tianbao Yang <tianbao-yang@tamu.edu>.

38th Conference on Neural Information Processing Systems (NeurIPS 2024).

Table 1: Comparison with existing stochastic methods for solving DWC problems with non-asymptotic convergence guarantee. $^*$ The method SBCD is designed to solve a problem in the form of $\min_x\{\min_y \phi(x,y) - \min_z \psi(x,z)\}$ with a specific formulation of $\phi$ and $\psi$. However, the method and analysis can be generalized to solving non-smooth DWC problems.

| Method | Smoothness of $\phi, \psi$ | Complexity | Loops |
|---|---|---|---|
| SDCA [26] | $\phi$: Smooth | $\mathcal{O}(\epsilon^{-4})$ | Double |
| SSDC [39] | $\phi$ or $\psi$: $\nu$-Hölder continuous gradient | $\mathcal{O}(\epsilon^{-4/\nu})$ | Double |
| SBCD$^*$ [45] | Non-smooth | $\mathcal{O}(\epsilon^{-6})$ | Double |
| SMAG (ours) | Non-smooth | $\mathcal{O}(\epsilon^{-4})$ | Single |

sparsity-promoting regularizers [39], Boltzmann machines [26]) and applications of min-max optimization (e.g., adversarial learning [31, 22], distributional robust learning [8, 28], learning with non-decomposable loss [28]). In recent years, the scale of data and models significantly increased, leading to the demand of more efficient optimization methods. However, all existing stochastic methods for DWC optimization and non-smooth WCSC min-max optimization with state-of-the-art non-asymptotic convergence rate $\mathcal{O}(\epsilon^{-4})$ are double-loop. As a result, these methods are complex regarding the implementation and require extensive hyperparameter tuning. To close this gap, we propose a single-loop stochastic algorithm for DMax optimization and provide non-asymptotic convergence analysis to match the state-of-the-art non-asymptotic convergence rate.

The main challenges of designing a single-loop method for DMax optimization are threefold. 1) given the weakly-convex nature of the component functions, their difference $F(x)$ is not necessarily weakly-convex, resulting in a non-smooth non-convex optimization problem. 2) the component functions $\max_{y\in\mathcal{Y}} \phi(x,y)$ and $\max_{z\in\mathcal{Z}} \psi(x,z)$ require solving maximization subproblems, making unbiased estimations of their subgradients inaccessible. 3) existing work on non-smooth problems with DC or/and min-max structures heavily rely on inner loops to solve subproblems to a certain accuracy.

To address the first challenge, we apply Moreau envelope smoothing technique [24, 3] to the component functions individually and take their difference as a smooth approximation of the original objective. Inspired by existing work [32, 45], we show that solving the original DMax problem can be achieved by solving this smooth approximation. Consequently, the problem is transformed into a smooth problem with two layer of nested optimization structure, the Moreau envelope and the maximization from the min-max structure. In order to avoid inner-loop, we perform only one step of update for each of the nested optimization problems. Our analysis leverages the fast convergence of strongly convex/concave problems, proving that single-step updates are sufficient to achieve a state-of-the-art convergence rate. Although the Moreau envelope smoothing is not new for solving DC and min-max optimization [32, 45, 47, 43], the existing results either require double loops [32, 45] or require smoothness of the objective function [47, 43].

**Contributions.** We summarize the main contribution of this work as following.

- We construct a new framework DMax optimization that unifies the DWC optimization and WCSC min-max optimization. Based on a Moreau envelope smoothing technique, we propose a single-loop stochastic algorithm, namely SMAG, for DMax optimization in non-smooth setting, which achieves $\mathcal{O}(\epsilon^{-4})$ convergence rate.

- We show that the proposed method leads to the first single-loop stochastic algorithms for DWC optimization and non-smooth WCSC min-max optimization achieving $\mathcal{O}(\epsilon^{-4})$ convergence rate.

- Finally, we present experimental results on applications including Positive-Unlabeled (PU) Learning and partial AUC optimization with an adversarial fairness regularizer to validate the effectiveness of our proposed algorithms.

## 2 Related Work

**Stochastic DC Optimization.** DWC can be converted into Difference-of-convex (DC) programming. DC programming was initially introduced in [33] and has been extensively studied since then. A comprehensive review on the developments of DC programming can be found in [18]. Despite the

Table 2: Comparison with existing stochastic methods for solving non-convex non-smooth min-max problems. The objective function is in the form of $\phi(x,y) = f(x,y) - g(y) + h(x)$. NS and S stand for non-smooth and smooth respectively, and NSP means non-smooth and its proximal mapping is easily solved. WC, C stand for weakly-convex and convex respectively. WCSC stands for weakly-convex-strongly-concave, SSC stands for smooth and strongly concave and WCC means weakly-convex-concave. Note that Epoch-GDA and SMAG studies the general formulation $\phi(x,y) = f(x,y)$.

| Method | $f(x,y)$ | $g(y)$ | $h(x)$ | Complexity | Loops |
|---|---|---|---|---|---|
| PG-SMD [29] | NS, WCC | NSP, SC | NSP, C | $\mathcal{O}(\epsilon^{-4})$ | Double |
| SAPD+ [48] | SSC | NSP, C | NSP,C | $\mathcal{O}(\epsilon^{-4})$ | Double |
| Epoch-GDA [40] | NS, WCSC | - | - | $\mathcal{O}(\epsilon^{-4})$ | Double |
| StocAGDA [1] | SSC | NSP, C | NSP, C | $\mathcal{O}(\epsilon^{-4})$ | Single |
| SMAG (ours) | NS,WCSC | - | - | $\mathcal{O}(\epsilon^{-4})$ | Single |

rich literature on DC programming, DC in stochastic setting has rarely been mentioned until recently. Most of the existing studies on stochastic DC optimization are based on the classical method, DC Algorithm (DCA) in deterministic DC optimization. The main idea of DCA is to approximate the DC problem by a convex problem by taking the linear approximation of the second component. In other words, DCA solves $\min_x \{\phi(x) - \langle \nabla \phi(x_k), x \rangle\}$ to update $x_k$ and thus forms a double-loop algorithm. [34] first proposed stochastic DCA (SDCA) for solving large sum problems of non-convex smooth functions, which was further generalized to solving large sum non-smooth problems in [16]. [15] is the first work that allows both components in DC problems to be non-smooth. The authors proposed a SDCA scheme in the aggregated update style, where all past information needs to be stored for constructing future subproblems. [17] improved the efficiency of the SDCA scheme by removing the need of storing historical information. So far, none of the above work provides non-asymptotic convergence guarantee. The first non-asymptotic convergence analysis was established in [26]. The authors proposed a stochastic proximal DC algorithm (SPD), which modifies SDCA by adding an extra quadratic term after linearizing the second component function, and proved that SPD has a convergence rate of $\mathcal{O}(\epsilon^{-4})$. The main drawback of their analysis is that they need the smoothness assumption of the first component function. With very similar algorithm design, [39] managed to partially relax the smoothness assumption. Given at least one of the two component functions having $\nu$-Hölder continuous gradient, i.e., $\|\nabla f(x) - \nabla f(x')\| \le \|x - x'\|^\nu$ for all $x, x'$, they proved a convergence rate of $\mathcal{O}(\epsilon^{-4/\nu})$. In fact, the Hölder continuous gradient assumption is still fairly strong as some of the common non-smooth functions do not satisfy, for example the hinge loss function.

Recently, another approach to tackling the non-smoothness in DC problems has been considered. Following the smoothing technique in non-smooth weakly-convex optimization literature [3], [32, 25] constructed Moreau envelope smoothing approximations for both of the component functions respectively and established non-asymptotic convergence analysis under deterministic setting and the assumption that either one component function is smooth or the proximal-point subproblems can be solved exactly. Following a similar idea, [45] studied a problem in the form of $\min_x F(x) := \min_y \phi(x,y) - \min_z \psi(x,z)$, where $\phi$ and $\psi$ are in some specific formulations, and proposed a double-loop algorithm with $\mathcal{O}(\epsilon^{-6})$ convergence rate. Although the $\phi$ and $\psi$ are non-smooth, their analysis heavily relies on the properties in the given formulation, especially the structures in the dual variables $y, z$, thus is not trivial to generalize.

Note that none of the aforementioned work is able to solve the DMax problem, as they require unbiased stochastic gradient estimations of the two component functions, which are not accessible in DMax due to the presence of the maximization structure.

**Stochastic Non-smooth Weakly-Convex-Strongly-Concave Min-Max Optimization.** Stochastic WCSC min-max optimization has been an emerging topic in recent years. Most of the existing works focuses on the smooth setting, i.e., the objective is smooth [12, 19, 49, 43, 47, 43] or the stochastic gradient oracles are Lipschitz continuous [23, 42, 11, 21, 38]. To the best of our knowledge, [29] is the first work that considers non-smooth WCSC min-max problems. They considered a special structure where the maximization over $y$ given $x$ can be simply solved and it is solved with $O(1/\epsilon^2)$ times. They proposed a nested method Proximally Guided Stochastic Mirror Descent Method (PG-SMD) that achieves a convergence rate of $\mathcal{O}(\epsilon^{-4})$. Later, [40] further relaxed the assumption by removing the requirement of the special structure, and proved that their nested method Epoch-GDA

has a similar convergence rate of $\mathcal{O}(\epsilon^{-4})$. Another line of work studies a special case of the general non-smooth non-convex min-max optimization, where the objective is assumed to be composite, i.e., $\phi(x,y) = f(x,y) - g(y) + h(x)$, so that $f$ is smooth while $g, h$ are potentially non-smooth [1, 48]. Both works established $\mathcal{O}(\epsilon^{-4})$ convergence rate, and assume $f$ is smooth and strongly concave, $g$ and $h$ are convex but potentially non-smooth and their proximal mappings can be easily solved. However, none of them is applicable to the general non-smooth WCSC min-max optimization.

## 3 Preliminaries

**Notations.** For simplicity, we denote $\Phi(x) := \max_{y \in \mathcal{Y}} \phi(x,y)$, $\Psi(x) := \max_{z \in \mathcal{Z}} \psi(x,z)$, $y^*(\cdot) := \arg\max_{y \in \mathcal{Y}} \phi(\cdot, y)$, and $z^*(\cdot) := \arg\max_{z \in \mathcal{Z}} \psi(\cdot, z)$. We use $\|\cdot\|$ to denote the Euclidean norm of a vector and $P_\mathcal{C}(\cdot)$ to denote the Euclidean projection onto a closed set $\mathcal{C}$. We use the following definitions of general subgradient and subdifferential [3, 30].

**Definition 3.1** (subgradient and subdifferential). Consider a function $f : \mathbb{R}^d \to \mathbb{R} \cup \{\infty\}$ and a point $x$ with finite $f(x)$. A vector $v \in \mathbb{R}^d$ is a general subgradient of $f$ at $x$ if

$$f(y) \geq f(x) + \langle v, y - x \rangle + o(\|y - x\|) \quad \text{as } y \to x.$$

The subdifferential $\partial f(x)$ is the set of subgradients of $f$ at point $x$.

For simplicity, we abuse the notation $\partial f(x)$ to denote one subgradient from the corresponding subdifferential when no confusion could be caused. We use $\tilde{\partial} f(x)$ and $\tilde{\nabla} f(x)$ to represent an unbiased stochastic estimator of the subgradient $\partial f(x)$ and the gradient $\nabla f(x)$ respectively. A function $f : \mathcal{D} \to \mathbb{R}$ is said to be $L$-smooth if $\|\nabla f(x) - \nabla f(x')\| \leq L\|x - x'\|$ for all $x, x' \in \mathcal{D}$. A function $f : \mathbb{R}^d \to \mathbb{R} \cup \{\infty\}$ is $\delta$-weakly convex if $f(\cdot) + \frac{\delta}{2}\|\cdot\|^2$ is convex. A mapping $\mathcal{M} : \mathcal{D} \to \mathbb{R}^l$ is said to be $C$-Lipschitz continuous if $\|\mathcal{M}(x) - \mathcal{M}(x')\| \leq C\|x - x'\|$ for all $x, x' \in \mathcal{D}$.

Consider solving a non-smooth problem $\min_x f(x)$. One of the main challenges is that the $\epsilon$-stationary point, i.e., a point $x$ such that $\text{dist}(0, \partial f(x)) \leq \epsilon$, which is the typical goal for smooth problems, may not exist in the neighborhood of its optimal solution. A classical counter example would be $f(x) = |x|$, where for $\epsilon \in [0, 1)$ the only $\epsilon$-stationary point is the optimal solution $x = 0$. A standard solution to this issue in weakly-convex setting is to use a relaxed convergence criteria, that is to find a point no more than $\epsilon$ away from an $\epsilon$-stationary point. This is called a nearly $\epsilon$-stationary point, and is widely used in non-smooth weakly-convex optimization literature [4, 29, 41, 50, 51, 19]. In fact, finding a nearly $\epsilon$-stationary point for $f(x)$ can be achieved by finding an $\epsilon$-stationary point of $f_\gamma(x)$, the Moreau envelope of $f(x)$. Assume function $f$ is $\delta$-weakly-convex, then its Moreau envelope and proximal map are given by

$$f_\gamma(x) := \min_{x'} \left\{ f(x') + \frac{1}{2\gamma}\|x' - x\|^2 \right\}, \quad \text{prox}_{\gamma f}(x) := \arg\min_{x'} \left\{ f(x') + \frac{1}{2\gamma}\|x' - x\|^2 \right\}.$$

Existing work [3] has shown that with $\gamma \in (0, \delta^{-1})$ and $\hat{x} = \text{prox}_{\gamma f}(x)$, we have

$$\nabla f_\gamma(x) = \gamma^{-1}(x - \hat{x}), \quad f(\hat{x}) \leq f(x), \quad \text{dist}(0, \partial f(\hat{x})) \leq \|\nabla f_\gamma(x)\|.$$

Moreover, $\text{prox}_{\gamma f}(x)$ is $\frac{1}{1-\gamma\delta}$ - Lipschitz continuous [32].

Now we consider the DMax problem (1). By Danskin's Theorem, the weak convexity assumption of $\phi(\cdot, y)$ and $\psi(\cdot, z)$ naturally leads to the weak convexity of $\Phi(\cdot)$ and $\Psi(\cdot)$. Since the weak convexity assumption of component functions does not guarantee the weak convexity of their difference function $F(x)$, one may neither 1) use nearly $\epsilon$-stationary point of $F(x)$ as the convergence metric, nor 2) directly apply Moreau envelope smoothing technique to $F(x)$. To tackle the first issue, we follow the existing work [45] to use the following convergence metric for non-smooth DWC problems.

**Definition 3.2** (Definition 2 in [45]). Given $\epsilon > 0$, we say $x$ is a **nearly $\epsilon$-critical point** of $\min_x\{F(x) := \Phi(x) - \Psi(x)\}$ if there exist $v, x', x''$ such that $v \in \partial\Phi(x') - \partial\Psi(x'')$ and $\max\{\mathbb{E}\|v\|, \mathbb{E}\|x - x'\|, \mathbb{E}\|x - x''\|\} \leq \epsilon$.

To tackle the second issue, we take the Moreau envelope of $\Phi(\cdot)$ and $\Psi(\cdot)$ individually and define the smooth approximation of $F(x)$ as

$$F_\gamma(x) = \Phi_\gamma(x) - \Psi_\gamma(x). \tag{4}$$

The recent work [32] has proven that $F_\gamma(x)$ is indeed smooth.

**Algorithm 1** Stochastic Moreau Envelope Approximate Gradient Method (SMAG)

---

1: **Input** Initial points: $x_\phi^0, x_\psi^0, x_0, y_0, z_0$. Hyper-parameters: $\gamma, \eta_0, \eta_1$.

2:        Stochastic (sub)gradients: $\tilde\partial_x\phi, \tilde\nabla_y\phi, \tilde\partial_x\psi, \tilde\nabla_z\psi$.

3: **for** $t = 0, \ldots, T-1$ **do**

4:      $x_\phi^{t+1} = x_\phi^t - \eta_1(\tilde\partial_x\phi(x_\phi^t, y_t) + \frac{1}{\gamma}(x_\phi^t - x_t))$

5:      $y_{t+1} = P_{\mathcal{Y}}\big(y_t + \eta_1\tilde\nabla_y\phi(x_\phi^t, y_t)\big)$

6:      $x_\psi^{t+1} = x_\psi^t - \eta_1(\tilde\partial_x\psi(x_\psi^t, z_t) + \frac{1}{\gamma}(x_\psi^t - x_t))$

7:      $z_{t+1} = P_{\mathcal{Z}}\big(z_t + \eta_1\tilde\nabla_z\psi(x_\psi^t, z_t)\big)$

8:      $G_{t+1} = \frac{1}{\gamma}(x_t - x_\phi^{t+1}) - \frac{1}{\gamma}(x_t - x_\psi^{t+1})$

9:      $x_{t+1} = x_t - \eta_0 G_{t+1}$

10: **end for**

11: **return** $x_\phi^{\bar{t}}$ or $x_\psi^{\bar{t}}$ with $\bar{t}$ uniformly sampled from $\{1, \ldots, T\}$

---

**Proposition 3.3** (Proposition EC.1.2 in [32]). *Assume $\Phi(\cdot)$ and $\Psi(\cdot)$ are $\delta_\phi, \delta_\psi$-weakly convex respectively. Then $F_\gamma(x) = \Phi_\gamma(x) - \Psi_\gamma(x)$ is $L_F$-smooth, where $L_F = \frac{2}{\gamma - \gamma^2\min\{\delta_\psi, \delta_\phi\}}$.*

Moreover, one can show that a good approximate stationary point $x$ of $F_\gamma(\cdot)$ and a good approximation point $x'$ to the proximal points $\text{prox}_{\gamma\Phi}(x)$ and $\text{prox}_{\gamma\Psi}(x)$ can guarantee that $x'$ is a nearly $\epsilon$-critical point of $\min_{\hat{x}} F(\hat{x})$.

**Lemma 3.4** (Lemma 3 in [45]). *Assume $\Phi(\cdot)$ and $\Psi(\cdot)$ are $\delta_\phi, \delta_\psi$-weakly convex respectively, and $0 < \gamma < \min\{\delta_\phi^{-1}, \delta_\psi^{-1}\}$. If $x$ is a vector such that $\mathbb{E}[\|\nabla F_\gamma(x)\|^2] \leq \min\{1, \gamma^{-2}\}\epsilon^2/4$, and $x'$ is a vector such that $\mathbb{E}[\|x' - \text{prox}_{\gamma\Phi}(x)\|^2] \leq \epsilon^2/4$ or $\mathbb{E}[\|x' - \text{prox}_{\gamma\Psi}(x)\|^2] \leq \epsilon^2/4$, then $x'$ is a nearly $\epsilon$-critical point of $\min_{\hat{x}} F(\hat{x})$.*

## 4 Algorithms and Convergence

Since we aim to minimize the smooth function $F_\gamma(x)$, the natural strategy is to perform gradient descent to update the variable $x$. Following from the properties of Moreau envelope, the gradient of $F_\gamma(x)$ is given by

$$\nabla F_\gamma(x) = \frac{1}{\gamma}(x - \text{prox}_{\gamma\Phi}(x)) - \frac{1}{\gamma}(x - \text{prox}_{\gamma\Psi}(x)), \tag{5}$$

where the blue component is the gradient of $\Phi_\gamma(x)$ and the green component is the gradient of $\Psi_\gamma(x)$. However, the proximal points $\text{prox}_{\gamma\Psi}(x)$ and $\text{prox}_{\gamma\Phi}(x)$ are not accessible in general. Indeed, these proximal points are the optimal solutions to $\min_{x'}\{\Phi(x') + \frac{1}{2\gamma}\|x - x'\|^2\}$ and $\min_{x'}\{\Psi(x') + \frac{1}{2\gamma}\|x - x'\|^2\}$ respectively, and $\Phi(\cdot)$ and $\Psi(\cdot)$ are typically not accessible because they are the value functions of possibly sophisticated maximization problems. Thus, we maintain two variables $x_\phi^t$ and $x_\psi^t$ as the estimators of $\text{prox}_{\gamma\Phi}(x_t)$ and $\text{prox}_{\gamma\Psi}(x_t)$ respectively, and maintain another two variables $y_t$ and $z_t$ as the estimators of $\arg\max_{y\in\mathcal{Y}}\phi(\text{prox}_{\gamma\Phi}(x_t), y)$ and $\arg\max_{z\in\mathcal{Z}}\psi(\text{prox}_{\gamma\Psi}(x_t), z)$ respectively. At each iteration, we update $x_\phi^t$ and $x_\psi^t$ by one step of stochastic gradient descent, and update $y_t$ and $z_t$ by one step of stochastic gradient ascent. Finally, we compute the gradient estimator $G_{t+1} = \frac{1}{\gamma}(x_t - x_\phi^{t+1}) - \frac{1}{\gamma}(x_t - x_\psi^{t+1})$ of $\nabla F_\gamma(x_t)$ and update $x_t$ by one step of gradient descent. The resulting algorithm is presented in Algorithm 1.

**DWC Optimization.** For DWC problem (2), the associated functions $\Phi(\cdot) = \phi(\cdot)$ and $\Psi(\cdot) = \psi(\cdot)$ are directly accessible. Thus the variables $y_t$ and $z_t$ in SMAG are no longer needed. The simplified SMAG algorithm for DWC optimization is presented in Algorithm 2.

**WCSC Min-Max Optimization.** For WCSC Min-Max problem (3), the second component function $\Psi = 0$ can be ignored, and thus variables $x_\psi^t$ and $z_t$ are no longer needed. However, this brings a

change to the gradient of $F_\gamma(x_t)$ as it now becomes

$$\nabla F_\gamma(x_t) = \gamma^{-1}(x_t - \text{prox}_{\gamma\Phi}(x_t)).$$

The simplified SMAG algorithm for WCSC Min-Max optimization is presented in Algorithm 3.

---

**Algorithm 2** SMAG for DWC Optimization

1: **for** $t = 0, \ldots, T-1$ **do**
2:     $x_\phi^{t+1} = x_\phi^t - \eta_1(\tilde{\partial}_x\phi(x_\phi^t) + \frac{1}{\gamma}(x_\phi^t - x_t))$
3:     $x_\psi^{t+1} = x_\psi^t - \eta_1(\tilde{\partial}_x\psi(x_\psi^t) + \frac{1}{\gamma}(x_\psi^t - x_t))$
4:     $G_{t+1} = \frac{1}{\gamma}(x_\psi^{t+1} - x_\phi^{t+1})$
5:     $x_{t+1} = x_t - \eta_0 G_{t+1}$
6: **end for**
7: **return** $x_\phi^{\bar{t}}$ or $x_\psi^{\bar{t}}$ with $\bar{t} \sim \{1, \ldots, T\}$

---

**Algorithm 3** SMAG for WCSC Min-Max Optimization

1: **for** $t = 0, \ldots, T-1$ **do**
2:     $x_\phi^{t+1} = x_\phi^t - \eta_1(\tilde{\partial}_x\phi(x_\phi^t, y_t) + \frac{1}{\gamma}(x_\phi^t - x_t))$
3:     $y_{t+1} = P_{\mathcal{Y}}(y_t + \eta_1\tilde{\nabla}_y\phi(x_\phi^t, y_t))$
4:     $G_{t+1} = \frac{1}{\gamma}(x_t - x_\phi^{t+1})$
5:     $x_{t+1} = x_t - \eta_0 G_{t+1}$
6: **end for**
7: **return** $x_{\bar{t}}$ with $\bar{t} \sim \{0, \ldots, T-1\}$

---

### 4.1 Convergence Analysis

In this section, we present convergence results for Algorithms 1-3. To proceed, we make the following assumption for DMax problem (1).

**Assumption 4.1.** Considering DMax problem (1), we assume that

(i) $\phi(\cdot, y)$ is $\delta_\phi$-weakly convex, and $\psi(\cdot, z)$ is $\delta_\psi$-weakly convex.

(ii) $\phi(x, \cdot)$ is $\mu_\phi$-strongly concave, and $\psi(x, \cdot)$ is $\mu_\psi$-strongly concave.

(iii) $\phi(x, y)$ and $\psi(x, z)$ are differentiable in terms of $y$ and $z$ respectively, $\nabla_y\phi(\cdot, y)$ is $L_{\phi,yx}$-Lipschitz continuous, and $\nabla_z\psi(\cdot, z)$ is $L_{\psi,zx}$-Lipschitz continuous.

(iv) There exists a constant $F_\gamma^* > -\infty$ such that $F_\gamma^* \le F_\gamma(x)$ for all $x$.

(v) There exists a finite constant $M$ such that $\mathbb{E}\|\tilde{\partial}_x\phi(x, y)\|^2 \le M^2$, $\mathbb{E}\|\tilde{\nabla}_y\phi(x, y)\|^2 \le M^2$, $\mathbb{E}\|\tilde{\partial}_x\psi(x, z)\|^2 \le M^2$, $\mathbb{E}\|\tilde{\nabla}_z\psi(x, z)\|^2 \le M^2$ for all $x \in \mathbb{R}^{d_x}$, $y \in \mathcal{Y}$ and $z \in \mathcal{Z}$.

It shall be noted that Assumption 4.1(iii) only requires partial smoothness of $\phi$ and $\psi$, and is to ensure the Lipschitz continuity of $y^*(\cdot) := \arg\max_{y\in\mathcal{Y}}\phi(\cdot, y)$ and $z^*(\cdot) := \arg\max_{z\in\mathcal{Z}}\psi(\cdot, z)$. This follows from existing results.

**Lemma 4.2** (Lemma 4.3 in [19]). *Consider problem* $\max_{y\in\hat{\mathcal{Y}}} f(x, y)$ *for any* $x \in \mathbb{R}^{d_x}$, *where* $\hat{\mathcal{Y}} \subset \mathbb{R}^{d_y}$ *is a closed convex set. Assume that* $f(x, y)$ *is* $\mu$-*strongly concave in* $y$ *for each* $x \in \mathbb{R}^{d_x}$, *and* $\nabla_y f(\cdot, y)$ *is* $L_{yx}$-*Lipschitz for each* $y \in \hat{\mathcal{Y}}$. *Then* $\arg\max_y f(\cdot, y)$ *is* $\frac{L_{yx}}{\mu}$-*Lipschitz continuous.*

A Lipschitz smooth function $f(x, y)$ is guaranteed to have Lipschitz continuous partial gradient $\nabla_y f(\cdot, y)$, while the reverse statement is not necessarily true. For example, consider a function $f(x, y) = y^\top h(x) - g(y)$ with non-smooth $C$-Lipschitz continuous $h(\cdot)$ and strongly convex $g$. Then $f(x, y)$ is non-smooth but the partial subgradient $\nabla_y f(\cdot, y) = h(\cdot) - \nabla g(y)$ is Lipschitz continuous with respect to the first argument. Another example is given by $f(x, y) = f_1(x) + f_2(x, y)$, where $f_1$ is weakly convex and $f_2$ is smooth and strongly concave in terms of $y$. The latter is indeed seen in our considered application for pAUC maximization with adversarial fairness. In fact, one may replace Assumption 4.1(iii) by directly assuming that $y^*(\cdot)$ and $z^*(\cdot)$ are Lipschitz continuous. In addition, Assumption 4.1(v) is standard in non-smooth optimization literature [3, 39, 10].

Here we give a brief outline of the convergence analysis. First of all, we present a standard result [7].

**Lemma 4.3.** *Suppose that* $F_\gamma(\cdot)$ *is* $L_F$-*smooth and* $x_{t+1} = x_t - \eta_0 G_{t+1}$ *with* $0 < \eta_0 \le \frac{1}{2L_F}$. *Then we have*

$$F_\gamma(x_{t+1}) \le F_\gamma(x_t) + \frac{\eta_0}{2}\|\nabla F_\gamma(x_t) - G_{t+1}\|^2 - \frac{\eta_0}{2}\|\nabla F_\gamma(x_t)\|^2 - \frac{\eta_0}{4}\|G_{t+1}\|^2.$$

This implies that the key to bounding the gradient $\|\nabla F_\gamma(x_t)\|^2$ is to obtain a recursive bound for the gradient estimation error $\|\nabla F_\gamma(x_t) - G_{t+1}\|^2$. Following from the true gradient formulation 5, we have

$$\|\nabla F_\gamma(x_t) - G_{t+1}\|^2 \leq \frac{2}{\gamma^2} \left( \|x_\phi^{t+1} - \text{prox}_{\gamma\Phi}(x_t)\|^2 + \|x_\psi^{t+1} - \text{prox}_{\gamma\Psi}(x_t)\|^2 \right). \quad (6)$$

In other words, the error of the gradient estimation $G_{t+1}$ can be bounded by the estimation errors of $x_\phi^{t+1}$ and $x_\psi^{t+1}$. Thus, we construct recursive bound for the proximal point estimation errors $\|x_\phi^{t+1} - \text{prox}_{\gamma\Phi}(x_t)\|^2$ and $\|x_\psi^{t+1} - \text{prox}_{\gamma\Psi}(x_t)\|^2$ individually. In fact, these two errors share almost identical analysis due to similar assumptions and updates. Here we only present the result for function $\phi$, as the result for $\psi$ directly follows.

**Lemma 4.4.** *Suppose that Assumption 4.1 holds, $0 < \gamma < 1/\delta_\phi$, and $\eta_1 \leq \frac{\gamma^2(1/\gamma - \delta_\phi)}{2}$. Then the sequences $\{x_t\}$, $\{y_t\}$, $\{x_\phi^t\}$ and $\{G_t\}$ generated by Algorithm 1 satisfy*

$$\mathbb{E}\|x_\phi^{t+1} - prox_{\gamma\Phi}(x_t)\|^2 + \mathbb{E}_t\|y_{t+1} - y^*(prox_{\gamma\Phi}(x_t))\|^2$$

$$\leq (1 - \frac{\eta_1(1/\gamma - \delta_\phi)}{2})\mathbb{E}\|x_\phi^t - prox_{\gamma\Phi}(x_{t-1})\|^2 + (1 - \eta_1\mu_\phi)\mathbb{E}\|y_t - y^*(prox_{\gamma\Phi}(x_{t-1}))\|^2$$

$$+ \left( \frac{2\eta_0^2}{\eta_1\gamma^2(1/\gamma - \delta_\phi)^3} + \frac{L_{\phi,yx}^2\eta_0^2}{\eta_1\mu_\phi^3\gamma^2(1/\gamma - \delta_\phi)^2} \right) \mathbb{E}\|G_t\|^2 + 12M^2\eta_1^2.$$

Finally, combining Lemma 4.3, inequality (6) and Lemma 4.4 yields the following convergence result for Algorithm 1.

**Theorem 4.5.** *Suppose that Assumption 4.1 holds, $0 < \gamma < \min\{\delta_\phi^{-1}, \delta_\psi^{-1}\}$, $\eta_1 = \mathcal{O}(\epsilon^2)$, and $\eta_0 = \tau\eta_1$. Then after $T \geq \mathcal{O}(\epsilon^{-4})$ iterations, the sequences $\{x_t\}$, $\{x_\phi^t\}$ and $\{x_\psi^t\}$ generated by Algorithm 1 satisfy $\mathbb{E}[\|x_\phi^{\bar{t}} - prox_{\gamma\Phi}(x_{\bar{t}-1})\|^2 + \|x_\psi^{\bar{t}} - prox_{\gamma\Psi}(x_{\bar{t}-1})\|^2 + \|\nabla F_\gamma(x_{\bar{t}-1})\|^2] \leq \min\{1, \gamma^{-2}\}\epsilon^2/4$, and the outputs $x_\phi^{\bar{t}}$ and $x_\psi^{\bar{t}}$ are both nearly $\epsilon$-critical points of problem (1).*

Since DMax optimization is a unified framework covering DWC optimization and WCSC min-max optimization, the convergence results of Algorithms 2 and 3 directly follow from Theorem 4.5. To present them, we first provide a reduced version of Assumption 4.1 for DWC problem (2).

**Assumption 4.6.** Considering DWC problem (2), we assume that

(i) $\phi(\cdot)$ is $\delta_\phi$-weakly convex, and $\psi(\cdot)$ is $\delta_\psi$-weakly convex.

(ii) There exists a constant $F_\gamma^* > -\infty$ such that $F_\gamma^* \leq F_\gamma(x)$ for all $x$.

(iii) There exists a finite constant $M$ such that $\mathbb{E}\|\tilde{\partial}\phi(x)\|^2 \leq M^2$ and $\mathbb{E}\|\tilde{\partial}\psi(x)\|^2 \leq M^2$ for all $x \in \mathbb{R}^{d_x}$.

By setting $\phi(x, y) = \phi(x)$ and $\psi(x, z) = \psi(x)$, namely independent of $y$ and $z$, in DMax problem (1), we obtain the following convergence result for Algorithm 2, which is an immediate consequence of Theorem 4.5.

**Corollary 4.7.** *Suppose that Assumption 4.6 holds, $0 < \gamma < \min\{\delta_\phi^{-1}, \delta_\psi^{-1}\}$, $\eta_1 = \mathcal{O}(\epsilon^2)$, and $\eta_0 = \tau\eta_1$. Then after $T \geq \mathcal{O}(\epsilon^{-4})$ iterations, the outputs $x_\phi^{\bar{t}}$ and $x_\psi^{\bar{t}}$ of Algorithm 2 are both nearly $\epsilon$-critical points of problem (2).*

For WCSC min-max problem (3), we reduce Assumption 4.1 to the following.

**Assumption 4.8.** Considering WCSC min-max problem (3), we assume that

(i) $\phi(\cdot, y)$ is $\delta_\phi$-weakly convex, and $\phi(x, \cdot)$ is $\mu_\phi$-strongly concave.

(ii) $\phi(x, y)$ is differentiable in terms of $y$, and $\nabla_y\phi(\cdot, y)$ is $L_{\phi,yx}$-Lipschitz continuous.

(iii) There exists a constant $F_\gamma^* > -\infty$ such that $F_\gamma^* \leq F_\gamma(x)$ for all $x$.

(iv) There exists a finite constant $M$ such that $\mathbb{E}\|\tilde{\partial}_x\phi(x, y)\|^2 \leq M^2$ and $\mathbb{E}\|\tilde{\nabla}_y\phi(x, y)\|^2 \leq M^2$ for all $x \in \mathbb{R}^{d_x}$ and $y \in \mathcal{Y}$.

By setting $\psi(x, z) = 0$ in DMax problem (1), we obtain the following convergence result for Algorithm 3, which is an immediate consequence of Theorem 4.5.

**Corollary 4.9.** *Suppose that Assumption 4.8 holds, $0 < \gamma < 1/\delta_\phi$, $\eta_1 = \mathcal{O}(\epsilon^2)$, and $\eta_0 = \tau\eta_1$, Then after $T \geq \mathcal{O}(\epsilon^{-4})$ iterations, the output $x_{\bar{t}}$ of Algorithm 3 is a nearly $\epsilon$-stationary point of problem (3).*

It shall be mentioned that for WCSC min-max problem (3, we use nearly $\epsilon$-stationary point as the convergence metric. This is standard in weakly-convex optimization literature [3].

# 5 Applications

In this section, we introduce two applications of DMax optimization, PU learning for DWC optimization and partial AUC optimization with adversarial fairness regularization for WCSC min-max optimization. We also show experimental results on both applications.

## 5.1 Positive-Unlabeled Learning

In binary classification task, the optimization problem is commonly formulated as the minimization of empirical risk, i.e., $\min_{\mathbf{w} \in \mathbb{R}^d} \frac{1}{|\mathcal{S}|} \sum_{\mathbf{x}_i \in \mathcal{S}} \ell(\mathbf{w}; \mathbf{x}_i, y_i)$ where $\ell(\mathbf{w}; \mathbf{x}_i, y_i)$ is the loss given the model parameter $\mathbf{w}$ on a data point $\mathbf{x}_i$ and its ground truth label $y_i$. Given the scenario where only positive data $\mathcal{S}_+$ are observed, then the standard approach becomes problematic. One way to address this issue is to utilize unlabeled data $\mathcal{S}_u$ to construct unbiased risk estimators. To be specific, [13] formulated the PU learning problem as following

$$\min_{\mathbf{w} \in \mathbb{R}^d} \frac{\pi_p}{n_+} \sum_{\mathbf{x}_i \in \mathcal{S}_+} [\ell(\mathbf{w}; \mathbf{x}_i, +1) - \ell(\mathbf{w}; \mathbf{x}_i, -1)] + \frac{1}{n_u} \sum_{\mathbf{x}_j^u \in \mathcal{S}_u} \ell(\mathbf{w}; x_j^u, -1) \tag{7}$$

where $n_+ = |\mathcal{S}_+|$, $n_u = |\mathcal{S}_u|$, $\pi_p = Pr(y = 1)$ is the prior probability of the positive class. If $\ell(\mathbf{w}; \mathbf{x}, y)$ is weakly convex in terms of $\mathbf{w}$, then Problem (7) is a DWC problems. In particular, in our experiments we consider linear classification model and hinge loss.

**Baselines.** We implemented five baselines and compared them with our proposed method SMAG for DWC optimization. The first baseline, stochastic gradient descent (SGD), does not have theoretical convergence guarantee for DWC problems. However, since it is the fundamental method for convex optimization, we include it to show its performance. We also implemented existing stochastic methods for solving DC or DWC problems with non-smooth components, including SDCA [26], SSDC-SPG [39], SSDC-Adagrad [39] and SBCD [45].

**Datasets.** We use four multi-class classification datasets, Fashion-MNIST [36], MNIST [5] CIFAR10 [14] and FER2013 [6]. To fit them in binary classification task, we consider the first five classes as negative for Fashion-MNIST, MNIST and CIFAR10, and the first four classes as negative for FER2013. For Fashion-MNIST, MNIST, CIFAR10, we follow the standard train-test split. For FER2013, we take the first 25709 samples as the training data, and the rest as for testing.

**Setup.** For all datasets, we use a batch size of 64 and set $\pi_p = 0.5$. We train 40 epochs and decay the learning rate by 10 at epoch 12 and 24. The learning rates of SGD, SDCA, SSDC-SPG and SSDC-Adagrad, the learning rate of the inner loop of SBCD (i.e., $\mu\eta_t/(\mu + \eta_t)$), and $\eta_1$ in SMAG are all tuned from $\{10, 1, 0.2, 0.1, 0.01, 0.001\}$. The learning rate of the outer loop in SDCA and $\eta_0$ in SMAG are tuned from $\{0.1, 0.5, 0.9\}$. The numbers of inner loops for all double-loop methods are tuned from $\{2, 5, 10\}$. The $\mu$ in SBCD, $1/\gamma$ in SSDC-SPG and SSDC-Adagrad, $\gamma$ in SMAG are tuned in $\{0.05, 0.1, 0.2, 0.5, 1, 2\}$. We run 4 trails for each setting and plot the average curves.

**Results.** We plot the curves of training losses in Figure 1. For all tested datasets, the performance of SMAG surpasses the baselines. Among the baselines, SBDC is the generally the next best choice. However, since SBDC is a double-loop method, it has one more hyperparameter compared to SMAG. We also present the ablation study of SMAG regarding the parameter $\gamma$ in Figure 2 included in the Appendix.

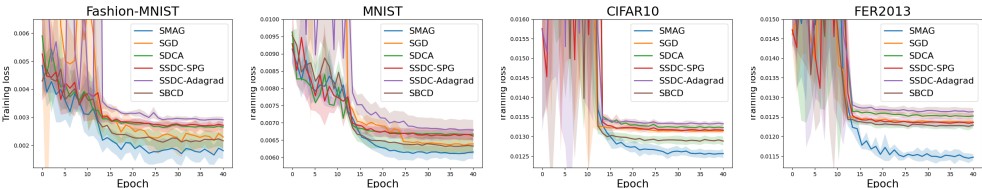

Figure 1: Training Curves of PU Learning

## 5.2 Partial AUC Maximization with Fairness Regularization

AUC Maximization aims to maximize the area under the curve of true positive rate (TPR) vs false positive rate (FPR). It has been studied extensively [44, 46, 20, 9] and has shown great success in large-scale real-world tasks, e.g., medical image classification [46] and molecular properties prediction [35]. One-way partial AUC (OPAUC) is an extension of AUC that has a primary interest in the curve corresponding to low FPR. To be specific, OPAUC restrict the FPR to the region $[0, \rho]$ where $\rho \in (0, 1)$. A recent work [52] proposed to formulate OPAUC problem into a non-smooth weakly convex optimization problem using conditional-value-at-risk (CVaR) based distributionally robust optimization (DRO). The formulation is given by

$$\min_{\mathbf{w}, \mathbf{s} \in \mathbb{R}^{n_+}} F_{\text{pauc}}(\mathbf{w}, \mathbf{s}) = \frac{1}{n_+} \sum_{\mathbf{x}_i \in \mathcal{S}_+} \left( s_i + \frac{1}{\rho n_-} \sum_{\mathbf{x}_j \in \mathcal{S}_-} (L(\mathbf{w}; \mathbf{x}_i, \mathbf{x}_j) - s_i)_+ \right), \tag{8}$$

where $\mathcal{S}_+, \mathcal{S}_-$ are the sets of positive and negative samples respectively, $n_+ = |\mathcal{S}_+|$, $n_- = |\mathcal{S}_-|$, and $\mathbf{w}$ denotes the weights of encoder network and classification layer. The pairwise surrogate loss is defined by $L(\mathbf{w}; \mathbf{x}_i, \mathbf{x}_j) = \ell(h(\mathbf{w}, \mathbf{x}_i) - h(\mathbf{w}, \mathbf{x}_j))$ and we use squared hinge loss as the surrogate loss, i.e., $\ell(\cdot) = (c - \cdot)^2$, where $c > 0$ is a parameter.

However, directly solving the above problem may end up with a model that is unfair with respect to some protected groups (e.g., female patients). Hence, we consider a formulation that incorporates an adversarial fairness regularization:

$$\max_{\mathbf{w}_a} F_{\text{fair}}(\mathbf{w}, \mathbf{w}_a) := \mathbb{E}_{(\mathbf{x}, a) \sim \mathcal{D}_a} \left\{ \mathbb{I}(a = 1) \log(\sigma(\mathbf{w}, \mathbf{w}_a, \mathbf{x})) + \mathbb{I}(a = -1) \log(1 - \sigma(\mathbf{w}, \mathbf{w}_a, \mathbf{x})) \right\},$$

where $\sigma(\mathbf{w}, \mathbf{w}_a, \mathbf{x})$ denotes a predicted probability that the data has a sensitive attribute $a = 1$ by using a classification head $\mathbf{w}_a$ on top of the encoded representation of $\mathbf{x}$. This adversarial fairness regularization has been demonstrated effective for promoting fairness [37]. As a result, we consider OPAUC problem with a fairness regularization:

$$\min_{\mathbf{w}, \mathbf{s} \in \mathbb{R}^{n_+}} \max_{\mathbf{w}_a} F_{\text{pauc}}(\mathbf{w}, \mathbf{s}) + \alpha F_{\text{fair}}(\mathbf{w}, \mathbf{w}_a) + \frac{\lambda_0}{2} \|\mathbf{w}_a\|_2^2 \tag{9}$$

It is clear that the problem is WCSC.

**Baseline.** We implement our proposed method SMAG for solving OPAUC problem (8) and OPAUC problem with adversarial fairness regularization (9). We refer the former as SMAG* and the latter as SMAG. The baseline on OPAUC problem (8) is SOPA, proposed in [52]. The baselines on OPAUC problem with adversarial fairness regularization (9) are SGDA [19] and Epoch-GDA [40].

**Dataset.** CelebA contains 200k celebrity face images with 40 binary attributes each, including the gender-sensitive attribute denoted as *Male*. In our experiments, we conduct experiments on three independent attribute prediction tasks: *Attractive, Big Nose, and Bags Under Eyes*, which have high Pearson correlations [2, 27] with the sensitive attribute *Male*. We divide the dataset into training, validation, and test data with an 80%/10%/10% split.

**Setup.** For all experiments, we adopt ResNet-18 as our backbone model architecture and initialize it with ImageNet pre-trained weights. The batch size is 128. We set the FPR upper bound to be $\rho = 0.3$. We train the model for 3 epochs with cosine decay learning rates for all baselines. The regularizer parameter $\alpha$ is tuned in $0.1, 0.2, 0.5$ for SGDA, Epoch-GDA, and SMAG, and the adversarial learning rates are tuned in $0.001, 0.01, 0.1$. $\alpha = 0$ for SOPA and SMAG*. The initial learning rates for optimizing $\mathbf{w}$ are tuned in $0.1, 0.01, 0.001$ for all methods, while the weight interpolation parameters,

i.e., $\gamma$ in Epoch-GDA and SMAG, are also tuned in $0.1, 0.01, 0.001$. The inner loop step is tuned in $\{5, 10, 15\}$ for Epoch-GDA. $\eta_1$ in SMAG are tuned from $\{10, 1, 0.2, 0.1, 0.01, 0.001\}$.

**Results.** We report the experimental results on three fairness metrics [27], equalized odds difference (EOD), equalized opportunity (EOP), and demographic disparity (DP) in Table 3. We observe that SMAG consistently achieves the highest pAUC score and lowest disparities metrics across all tasks compared to all other baseline min-max methods.

Table 3: Mean $\pm$ std of fairness results on CelebA test dataset with *Attractive and Big Nose* task labels, and *Male* sensitive attribute. Results are reported on 3 independent runs. We use bold font to denote the best result and use underline to denote the second best. Results on *Bags Under Eyes* are included in the appendix due to limited space.

| Methods | Attractive, Male | | | | Big Nose, Male | | | |
|---|---|---|---|---|---|---|---|---|
| | pAUC↑ | EOD↓ | EOP↓ | DP↓ | pAUC↑ | EOD↓ | EOP↓ | DP↓ |
| SOPA | $0.8485 \pm 0.012$ | $0.2638 \pm 0.035$ | $0.2438 \pm 0.032$ | $0.4753 \pm 0.023$ | $0.8039 \pm 0.005$ | $0.2829 \pm 0.024$ | $0.2269 \pm 0.019$ | $0.4424 \pm 0.034$ |
| SMAG* | $\mathbf{0.8606} \pm 0.003$ | $\underline{0.2192} \pm 0.020$ | $0.2333 \pm 0.068$ | $0.4510 \pm 0.027$ | $\mathbf{0.8078} \pm 0.002$ | $\underline{0.2735} \pm 0.012$ | $\underline{0.2205} \pm 0.030$ | $\underline{0.4364} \pm 0.019$ |
| SGDA | $0.8509 \pm 0.001$ | $0.2701 \pm 0.020$ | $0.2549 \pm 0.025$ | $0.4860 \pm 0.015$ | $0.8038 \pm 0.002$ | $0.2846 \pm 0.023$ | $0.2398 \pm 0.029$ | $0.4390 \pm 0.028$ |
| EGDA | $0.8546 \pm 0.004$ | $0.2290 \pm 0.006$ | $\underline{0.1735} \pm 0.059$ | $\underline{0.4305} \pm 0.032$ | $0.8023 \pm 0.005$ | $0.3293 \pm 0.027$ | $0.3076 \pm 0.012$ | $0.4620 \pm 0.031$ |
| SMAG | $\underline{0.8605} \pm 0.002$ | $\mathbf{0.1900} \pm 0.023$ | $\mathbf{0.1648} \pm 0.064$ | $\mathbf{0.4116} \pm 0.031$ | $\underline{0.8058} \pm 0.001$ | $\mathbf{0.2708} \pm 0.021$ | $\mathbf{0.2148} \pm 0.021$ | $\mathbf{0.4333} \pm 0.013$ |

# 6   Conclusion

In this study, we have introduced a new framework namely DMax optimization, that unifies DWC optimization and non-smooth WCSC min-max optimization. We proposed a single-loop stochastic method for solving DMax optimization and presented a novel convergence analysis showing that the proposed method achieves a non-asymptotic convergence rate of $\mathcal{O}(\epsilon^{-4})$. Experimental results on two applications, PU learning and OPAUC optimization with adversarial fairness regularization demonstrate strong performance of our method. One limitation of this work is the strong convexity assumption on the $\phi(x, \cdot)$ and $\psi(x, \cdot)$. This strong assumption may limit the applicability of our method. Future work will focus on exploring DMax optimization with weaker assumptions.

# Acknowledgment

We thank anonymous reviewers for constructive comments. Q. Hu and T. Yang were partially supported by the National Science Foundation Career Award 2246753, the National Science Foundation Award 2246757, 2246756 and 2306572. Z. Lu was partially supported by the National Science Foundation Award IIS-2211491, the Office of Naval Research Award N00014-24-1-2702, and the Air Force Office of Scientific Research Award FA9550-24-1-0343.

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

## A  Convergence Analysis

Recall that $\Phi(x) := \max_{y \in \mathcal{Y}} \phi(x,y)$, $\Psi(x) := \max_{z \in \mathcal{Z}} \psi(x,z)$, $y^*(\cdot) := \arg\max_{y \in \mathcal{Y}} \phi(\cdot, y)$, and $z^*(\cdot) := \arg\max_{z \in \mathcal{Z}} \psi(\cdot, z)$. Before presenting the proof of Theorem 4.5, we first give the proof of the proximal point estimation error bounds. As we have stated the bound for $\|x_\phi^{t+1} - prox_{\gamma\Phi}(x_t)\|^2$ in Lemma 4.4, here we present the corresponding lemma for $\|x_\psi^{t+1} - prox_{\gamma\Psi}(x_t)\|^2$.

**Lemma A.1.** *Suppose that Assumption 4.1 holds, $0 < \gamma < 1/\delta_\psi$, and $\eta_1 \leq \frac{\gamma^2(1/\gamma - \delta_\psi)}{2}$. Then the sequences $\{x_t\}$, $\{z_t\}$, $\{x_\psi^t\}$ and $\{G_t\}$ generated by Algorithm 1 satisfy*

$$
\mathbb{E}\|x_\psi^{t+1} - prox_{\gamma\Psi}(x_t)\|^2 + \mathbb{E}\|z_{t+1} - z^*(prox_{\gamma\Psi}(x_t))\|^2
$$

$$
\leq (1 - \frac{\eta_1(1/\gamma - \delta_\psi)}{2})\mathbb{E}\|x_\psi^t - prox_{\gamma\Psi}(x_{t-1})\|^2 + (1 - \eta_1\mu_\psi)\mathbb{E}\|z_t - z^*(prox_{\gamma\Psi}(x_{t-1}))\|^2
$$

$$
+ \left( \frac{2\eta_0^2}{\eta_1\gamma^2(1/\gamma - \delta_\psi)^3} + \frac{L_{\psi,zx}^2\eta_0^2}{\eta_1\mu_\psi^3\gamma^2(1/\gamma - \delta_\psi)^2} \right) \mathbb{E}\|G_t\|^2 + 12M^2\eta_1^2
$$

Since Lemma 4.4 and Lemma A.1 share the same proof strategy, we only present the proof of Lemma 4.4.

### A.1  Proof of Lemma 4.4

*Proof.* Recall that $\Phi(x) = \max_{y \in \mathcal{Y}} \phi(x,y)$ and $y^*(\cdot) = \arg\max_{y \in \mathcal{Y}} \phi(\cdot, y)$. Observe from Assumption 4.1(i) that $\Phi$ is $\delta_\phi$-weakly convex. It then follows that $prox_{\gamma\Phi}(\cdot)$ is $1/(1 - \gamma\delta_\phi)$-Lipschitz continuous. By this, Assumption 4.1(iii) and Lemma 4.2, it is not hard to see that $y^*(prox_{\gamma\Phi}(\cdot))$ is $L_{\phi,yx}/(\mu_\phi(1 - \gamma\delta_\phi))$-Lipschitz continuous.

For notational convenience, we let

$$
\Phi_t(x,y) = \phi(x,y) + \frac{1}{2\gamma}\|x - x_t\|^2,
$$

$$
x_{\Phi,t}^* = prox_{\gamma\Phi}(x_t), \quad y_t^* = y^*(prox_{\gamma\Phi}(x_t)). \tag{10}
$$

In view of (10) and the update rule of $x_\phi^{t+1}$, one has

$$
\mathbb{E}_t\|x_\phi^{t+1} - x_{\Phi,t}^*\|^2 = \mathbb{E}_t\|x_\phi^t - \eta_1\tilde{\partial}_x\Phi_t(x_\phi^t, y_t) - x_{\Phi,t}^*\|^2
$$

$$
= \|x_\phi^t - x_{\Phi,t}^*\|^2 - 2\mathbb{E}_t\langle\eta_1\tilde{\partial}_x\Phi_t(x_\phi^t, y_t), x_\phi^t - x_{\Phi,t}^*\rangle + \mathbb{E}_t\|\eta_1\tilde{\partial}_x\Phi_t(x_\phi^t, y_t)\|^2
$$

$$
\leq \|x_\phi^t - x_{\Phi,t}^*\|^2 + 2\eta_1 \underbrace{\langle\partial_x\Phi_t(x_\phi^t, y_t), x_{\Phi,t}^* - x_\phi^t\rangle}_{(A)} + 8M^2\eta_1^2 + \frac{2\eta_1^2}{\gamma^2}\|x_\phi^t - x_{\Phi,t}^*\|^2, \tag{11}
$$

where we use the inequality

$$
\mathbb{E}_t\|\tilde{\partial}_x\Phi_t(x_\phi^t, y_t)\|^2 = \mathbb{E}_t\|\tilde{\partial}_x\phi(x_\psi^t, y_t) + \frac{1}{\gamma}(x_\phi^t - x_t)\|^2
$$

$$
= \mathbb{E}_t\|\tilde{\partial}_x\phi(x_\phi^t, y_t) + \frac{1}{\gamma}(x_\phi^t - x_t) - \partial_x\phi(x_{\Phi,t}^*, y_t^*) - \frac{1}{\gamma}(x_{\Phi,t}^* - x_t)\|^2
$$

$$
\leq 4\mathbb{E}_t\|\tilde{\partial}\phi(x_\phi^t, y_t)\|^2 + 4\|\partial_x\phi(x_{\Phi,t}^*, y_t^*)\|^2 + \frac{2}{\gamma^2}\|x_\phi^t - x_{\Phi,t}^*\|^2
$$

$$
\leq 8M^2 + \frac{2}{\gamma^2}\|x_\phi^t - x_{\Phi,t}^*\|^2.
$$

By $(\gamma^{-1} - \delta_\phi)$-strong convexity of $\Phi_t(\cdot, y)$ and the definition of $x_{\Phi,t}^*$ in (10), one has

$$
\langle\partial_x\Phi_t(x_\phi^t, y_t), x_{\Phi,t}^* - x_\phi^t\rangle \leq \Phi_t(x_{\Phi,t}^*, y_t) - \Phi_t(x_\phi^t, y_t) - \frac{(1/\gamma - \delta_\phi)}{2}\|x_{\Phi,t}^* - x_\phi^t\|^2,
$$

$$
0 \leq \Phi_t(x_\phi^t, y_t^*) - \Phi_t(x_{\Phi,t}^*, y_t^*) - \frac{(1/\gamma - \delta_\phi)}{2}\|x_{\Phi,t}^* - x_\phi^t\|^2.
$$

Summing up these two inequalities gives

$$(A) \leq \Phi_t(x^*_{\Phi,t}, y_t) - \Phi_t(x^t_\phi, y_t) + \Phi_t(x^t_\phi, y^*_t) - \Phi_t(x^*_{\Phi,t}, y^*_t) - (1/\gamma - \delta_\phi)\|x^*_{\Phi,t} - x^t_\phi\|^2. \quad (12)$$

Notice from the definition of $y^*_t$ in (10) that there exists a particular subgradient $\nabla_y \phi(x^*_{\Phi,t}, y^*_t)$ such that

$$y^*_t = P_{\mathcal{Y}}\big(y^*_t + \eta_1 \nabla_y \phi(x^*_{\Phi,t}, y^*_t)\big).$$

Using this and the update rule of $y_{t+1}$, we have

$$\begin{aligned}
\mathbb{E}_t\|y_{t+1} - y^*_t\|^2 &= \mathbb{E}_t\|P_{\mathcal{Y}}(y_t + \eta_1 \tilde{\nabla}_y \Phi(x^t_\phi, y_t)) - y^*_t\|^2 \\
&= \mathbb{E}_t\|P_{\mathcal{Y}}(y_t + \eta_1 \tilde{\nabla}_y \Phi_t(x^t_\phi, y_t)) - P_{\mathcal{Y}}(y^*_t + \eta_1 \nabla_y \Phi_t(x^*_{\Phi,t}, y^*_t))\|^2 \\
&\leq \mathbb{E}_t\|y_t + \eta_1 \tilde{\nabla}_y \Phi_t(x^t_\phi, y_t) - (y^*_t + \eta_1 \nabla_y \Phi_t(x^*_{\Phi,t}, y^*_t))\|^2 \\
&\leq \|y_t - y^*_t\|^2 + 2\eta_1 \langle \nabla_y \Phi_t(x^t_\phi, y_t) - \nabla_y \Phi_t(x^*_{\Phi,t}, y^*_t), y_t - y^*_t \rangle \\
&\quad + \eta_1^2 \mathbb{E}_t\|\tilde{\nabla}_y \Phi_t(x^t_\phi, y_t) - \nabla_y \Phi_t(x^*_{\Phi,t}, y^*_t)\|^2 \\
&\leq \|y_t - y^*_t\|^2 + 2\eta_1 \underbrace{\langle \nabla_y \Phi_t(x^t_\phi, y_t) - \nabla_y \Phi_t(x^*_{\Phi,t}, y^*_t), y_t - y^*_t \rangle}_{(B)} + 4\eta_1^2 M^2.
\end{aligned} \quad (13)$$

By $\mu_\phi$-strong concavity of $\Phi_t(x, \cdot)$, we have

$$\begin{aligned}
(B) &= \langle -\nabla_y \Phi_t(x^t_\phi, y_t), y^*_t - y_t \rangle + \langle -\nabla_y \Phi_t(x^*_{\Phi,t}, y^*_t), y_t - y^*_t \rangle \\
&\leq -\Phi_t(x^t_\phi, y^*_t) + \Phi_t(x^t_\phi, y_t) - \frac{\mu_\phi}{2}\|y^*_t - y_t\|^2 \\
&\quad - \Phi_t(x^*_{\Phi,t}, y_t) + \Phi_t(x^*_{\Phi,t}, y^*_t) - \frac{\mu_\phi}{2}\|y^*_t - y_t\|^2 \\
&= -\Phi_t(x^t_\phi, y^*_t) + \Phi_t(x^t_\phi, y_t) - \Phi_t(x^*_{\Phi,t}, y_t) + \Phi_t(x^*_{\Phi,t}, y^*_t) - \mu_\phi\|y^*_t - y_t\|^2.
\end{aligned} \quad (14)$$

Combining (12) and (14) yields

$$(A) + (B) \leq -(1/\gamma - \delta_\phi)\|x^*_{\Phi,t} - x^t_\phi\|^2 - \mu_\phi\|y^*_t - y_t\|^2.$$

Using this inequality, (11) and (13), we have

$$\begin{aligned}
&\mathbb{E}_t\|x^{t+1}_\phi - x^*_{\Phi,t}\|^2 + \mathbb{E}_t\|y_{t+1} - y^*_t\|^2 \\
&\leq (1 - 2\eta_1(1/\gamma - \delta_\phi) + 2\eta_1^2/\gamma^2)\|x^*_{\Phi,t} - x^t_\phi\|^2 + (1 - 2\eta_1\mu_\phi)\|y^*_t - y_t\|^2 + 12M^2\eta_1^2 \\
&\overset{(a)}{\leq} (1 - \eta_1(1/\gamma - \delta_\phi))\|x^*_{\Phi,t} - x^t_\phi\|^2 + (1 - 2\eta_1\mu_\phi)\|y^*_t - y_t\|^2 + 12M^2\eta_1^2 \\
&\overset{(b)}{\leq} (1 - \eta_1(1/\gamma - \delta_\phi))\left(\left(1 + \frac{\eta_1(1/\gamma - \delta_\phi)}{2}\right)\|x^t_\phi - x^*_{\Phi,t-1}\|^2 + \left(1 + \frac{2}{\eta_1(1/\gamma - \delta_\phi)}\right)\|x^*_{\Phi,t-1} - x^*_{\Phi,t}\|^2\right) \\
&\quad + (1 - 2\eta_1\mu_\phi)\left((1 + \eta_1\mu_\phi)\|y_t - y^*_{t-1}\|^2 + \left(1 + (\eta_1\mu_\phi)^{-1}\right)\|y^*_{t-1} - y^*_t\|^2\right) + 12M^2\eta_1^2 \\
&\overset{(c)}{\leq} \left(1 - \frac{\eta_1(1/\gamma - \delta_\phi)}{2}\right)\|x^t_\phi - x^*_{\Phi,t-1}\|^2 + \frac{2}{\eta_1(1/\gamma - \delta_\phi)}\|x^*_{\Phi,t-1} - x^*_{\Phi,t}\|^2 \\
&\quad + (1 - \eta_1\mu_\phi)\|y_t - y^*_{t-1}\|^2 + (\eta_1\mu_\phi)^{-1}\|y^*_{t-1} - y^*_t\|^2 + 12M^2\eta_1^2 \\
&\overset{(d)}{\leq} \left(1 - \frac{\eta_1(1/\gamma - \delta_\phi)}{2}\right)\|x^t_\phi - x^*_{\Phi,t-1}\|^2 + (1 - \eta_1\mu_\phi)\|y_t - y^*_{t-1}\|^2 \\
&\quad + \left(\frac{2\eta_0^2}{\eta_1\gamma^2(1/\gamma - \delta_\phi)^3} + \frac{L^2_{\phi,yx}\eta_0^2}{\eta_1\mu_\phi^3\gamma^2(1/\gamma - \delta_\phi)^2}\right)\|G_t\|^2 + 12M^2\eta_1^2,
\end{aligned}$$

where $(a)$ follows from the assumption $\eta_1 \leq \frac{\gamma^2(1/\gamma - \delta_\phi)}{2}$, $(b)$ uses the fact that $\|a + b\|^2 \leq (1 + \alpha)\|a\|^2 + (1 + \frac{1}{\alpha})\|b\|^2$ for any $\alpha > 0$, (c) follows from bounding the coefficient of each term from above, and $(d)$ uses $1/(1 - \gamma\delta_\phi)$-Lipschitz continuity of $\text{prox}_{\gamma\Phi}(\cdot)$, $L_{\phi,yx}/(\mu_\phi(1 - \gamma\delta_\phi))$-Lipschitz continuity of $y^*(\text{prox}_{\gamma\Phi}(\cdot))$ and the update rule of $x_t$.

Finally taking expectation over all randomness yields the desired result. $\qquad \square$

## A.2 Proof of Theorem 4.5

We first present a detailed version of Theorem 4.5.

**Theorem A.2.** *Consider Problem 1 and assume Assumption 4.1 holds. Suppose that the parameters $\gamma$, $\eta_0$ and $\eta_1$ in Algorithm 1 are chosen as follows:*

$$0 < \gamma < \min\{\delta_\phi^{-1}, \delta_\psi^{-1}\}, \quad \alpha = \min\left\{\frac{1/\gamma - \delta_\phi}{4}, \frac{1/\gamma - \delta_\psi}{4}, \mu_\phi, \mu_\psi\right\},$$

$$\tau = \min\left\{\frac{\gamma^2\alpha^2}{4}, \frac{\mu_\phi^{1.5}\gamma^2\alpha^{1.5}}{4L_{\phi,yx}}, \frac{\mu_\psi^{1.5}\gamma^2\alpha^{1.5}}{4L_{\psi,zx}}\right\}, \quad \nu = \min\left\{1, \frac{2\tau}{\gamma^2\alpha}\right\}, \quad L_F = \frac{2}{\gamma - \gamma^2\min\{\delta_\psi, \delta_\phi\}},$$

$$\eta_1 = \min\left\{\frac{\gamma^2(1/\gamma - \delta_\phi)}{2}, \frac{\gamma^2(1/\gamma - \delta_\psi)}{2}, \frac{1}{2L_F\tau}, \frac{\min\{1, \gamma^2\}\min\{\alpha, \tau\}\nu\alpha}{768\tau M^2}\epsilon^2\right\}, \quad \eta_0 = \tau\eta_1.$$

*Then we have*

$$\frac{1}{T}\sum_{t=0}^{T-1}\left(\mathbb{E}\|x_\phi^{t+1} - prox_{\gamma\Phi}(x_t)\|^2 + \mathbb{E}\|x_\psi^{t+1} - prox_{\gamma\Psi}(x_t)\|^2 + \mathbb{E}\|\nabla F_\gamma(x_t)\|^2\right) \leq \min\{1, \gamma^{-2}\}\frac{\epsilon^2}{4},$$

*and consequently $x_\phi^{\bar{t}}$ and $x_\psi^{\bar{t}}$ are both nearly $\epsilon$-critical points of problem* (1), *whenever*

$$T \geq \frac{16(F_\gamma(x_0) - F_\gamma^* + P_0)}{\min\{1, \gamma^{-2}\}\min\{\alpha, \tau\}\nu\epsilon^2}\max\left\{\frac{2}{\gamma^2(1/\gamma - \delta_\phi)}, \frac{2}{\gamma^2(1/\gamma - \delta_\psi)}, 2L_F\tau, \frac{768\tau M^2}{\min\{1, \gamma^2\}\min\{\alpha, \tau\}\nu\alpha\epsilon^2}\right\}$$

(15)

*with*

$$P_0 = \frac{2\eta_0}{\eta_1\gamma^2\alpha}\left(\mathbb{E}\|x_\phi^1 - prox_{\gamma\Phi}(x_0)\|^2 + \mathbb{E}\|y_1 - y_0^*\|^2 + \mathbb{E}\|x_\psi^1 - prox_{\gamma\Psi}(x_0)\|^2 + \mathbb{E}\|z_1 - z_0^*\|^2\right).$$

*Proof.* For notational convenience, let

$$x_{\Psi,t}^* = \text{prox}_{\gamma\Psi}(x_t), \quad z_t^* = \arg\max_{z \in \mathcal{Z}} \psi(x_{\Psi,t}^*, z). \tag{16}$$

From Proposition 3.3, we know that $F_\gamma(\cdot)$ is $L_F$-smooth. By this, $0 < \eta_0 \leq \frac{1}{2L_F}$, and Lemma 4.3, one has

$$F_\gamma(x_{t+1}) \leq F_\gamma(x_t) + \frac{\eta_0}{2}\|\nabla F_\gamma(x_t) - G_{t+1}\|^2 - \frac{\eta_0}{2}\|\nabla F_\gamma(x_t)\|^2 - \frac{\eta_0}{4}\|G_{t+1}\|^2. \tag{17}$$

Notice that

$$\nabla F_\gamma(x_t) = \gamma^{-1}(\text{prox}_{\gamma\Psi}(x_t) - x_t + x_t - \text{prox}_{\gamma\Phi}(x_t)) = \gamma^{-1}(\text{prox}_{\gamma\Psi}(x_t) - \text{prox}_{\gamma\Phi}(x_t)),$$

$$G_{t+1} = \gamma^{-1}(x_\psi^{t+1} - x_\phi^{t+1}).$$

Using these, (10) and (16), we have

$$\|\nabla F_\gamma(x_t) - G_{t+1}\|^2 = \|\gamma^{-1}(\text{prox}_{\gamma\Psi}(x_t) - \text{prox}_{\gamma\Phi}(x_t)) - \gamma^{-1}(x_\psi^{t+1} - x_\phi^{t+1})\|^2$$

$$= \|\gamma^{-1}(x_{\Psi,t}^* - x_{\Phi,t}^*) - \gamma^{-1}(x_\psi^{t+1} - x_\phi^{t+1})\|^2 \tag{18}$$

$$\leq 2\gamma^{-2}\left(\|x_{\Psi,t}^* - x_\psi^{t+1}\|^2 + \|x_{\Phi,t}^* - x_\phi^{t+1}\|^2\right).$$

It follows from this and (17) that

$$\mathbb{E}[F_\gamma(x_{t+1})] \leq \mathbb{E}[F_\gamma(x_t)] + \frac{\eta_0}{\gamma^2}\mathbb{E}\|x_\psi^{t+1} - x_{\Psi,t}^*\|^2 + \frac{\eta_0}{\gamma^2}\mathbb{E}\|x_\phi^{t+1} - x_{\Phi,t}^*\|^2$$

$$- \frac{\eta_0}{2}\mathbb{E}\|\nabla F_\gamma(x_t)\|^2 - \frac{\eta_0}{4}\mathbb{E}\|G_{t+1}\|^2. \tag{19}$$

Let $x_{\Phi,t}^*$ and $y_t^*$ be defined in (10). Invoking Lemma 4.4, we have

$$\mathbb{E}\|x_\phi^{t+2} - x_{\Phi,t+1}^*\|^2 + \mathbb{E}\|y_{t+2} - y_{t+1}^*\|^2$$

$$\leq \left(1 - \frac{\eta_1(1/\gamma - \delta_\phi)}{2}\right)\mathbb{E}\|x_\phi^{t+1} - x_{\Phi,t}^*\|^2 + (1 - \eta_1\mu_\phi)\mathbb{E}\|y_{t+1} - y_t^*\|^2$$

$$+ \left(\frac{2\eta_0^2}{\eta_1\gamma^2(1/\gamma - \delta_\phi)^3} + \frac{L_{\phi,yx}^2\eta_0^2}{\eta_1\mu_\phi^3\gamma^2(1/\gamma - \delta_\phi)^2}\right)\mathbb{E}\|G_{t+1}\|^2 + 12M^2\eta_1^2.$$

Recall that $x^*_{\Psi,t}$ and $z^*_t$ are defined in (16). By Lemma A.1, one has

$$
\mathbb{E}\|x^{t+2}_\psi - x^*_{\Psi,t+1}\|^2 + \mathbb{E}\|z_{t+2} - z^*_{t+1}\|^2
$$
$$
\leq \left(1 - \frac{\eta_1(1/\gamma - \delta_\psi)}{2}\right) \mathbb{E}\|x^{t+1}_\psi - x^*_{\Psi,t}\|^2 + (1 - \eta_1\mu_\psi)\mathbb{E}\|z_{t+1} - z^*_t\|^2
$$
$$
+ \left(\frac{2\eta_0^2}{\eta_1\gamma^2(1/\gamma - \delta_\psi)^3} + \frac{L^2_{\psi,zx}\eta_0^2}{\eta_1\mu_\psi^3\gamma^2(1/\gamma - \delta_\psi)^2}\right) \mathbb{E}\|G_{t+1}\|^2 + 12M^2\eta_1^2.
$$

Let $\alpha$ be given in the statement of this theorem. Using this and the last two inequalities above, we have

$$
\mathbb{E}\|x^{t+2}_\phi - x^*_{\Phi,t+1}\|^2 + \mathbb{E}\|y_{t+2} - y^*_{t+1}\|^2
$$
$$
\leq (1 - \alpha\eta_1)\left(\mathbb{E}\|x^{t+1}_\phi - x^*_{\Phi,t}\|^2 + \mathbb{E}\|y_{t+1} - y^*_t\|^2\right) \tag{20}
$$
$$
+ \left(\frac{2\eta_0^2}{\eta_1\gamma^2(1/\gamma - \delta_\phi)^3} + \frac{L^2_{\phi,yx}\eta_0^2}{\eta_1\mu_\phi^3\gamma^2(1/\gamma - \delta_\phi)^2}\right) \mathbb{E}\|G_{t+1}\|^2 + 12M^2\eta_1^2,
$$

$$
\mathbb{E}\|x^{t+2}_\psi - x^*_{\Psi,t+1}\|^2 + \mathbb{E}\|z_{t+2} - z^*_{t+1}\|^2
$$
$$
\leq (1 - \alpha\eta_1)\left(\mathbb{E}\|x^{t+1}_\psi - x^*_{\Psi,t}\|^2 + \mathbb{E}\|z_{t+1} - z^*_t\|^2\right) \tag{21}
$$
$$
+ \left(\frac{2\eta_0^2}{\eta_1\gamma^2(1/\gamma - \delta_\psi)^3} + \frac{L^2_{\psi,zx}\eta_0^2}{\eta_1\mu_\psi^3\gamma^2(1/\gamma - \delta_\psi)^2}\right) \mathbb{E}\|G_{t+1}\|^2 + 12M^2\eta_1^2.
$$

Summing up inequalities (19), (20)$\times \frac{2\eta_0}{\eta_1\gamma^2\alpha}$ and (21)$\times \frac{2\eta_0}{\eta_1\gamma^2\alpha}$ yields

$$
\mathbb{E}[F_\gamma(x_{t+1})] + \frac{2\eta_0}{\eta_1\gamma^2\alpha}\left(\mathbb{E}\|x^{t+2}_\phi - x^*_{\Phi,t+1}\|^2 + \mathbb{E}\|y_{t+2} - y^*_{t+1}\|^2\right)
$$
$$
+ \frac{2\eta_0}{\eta_1\gamma^2\alpha}\left(\mathbb{E}\|x^{t+2}_\psi - x^*_{\Psi,t+1}\|^2 + \mathbb{E}\|z_{t+2} - z^*_{t+1}\|^2\right)
$$
$$
\leq \mathbb{E}[F_\gamma(x_t)] + \frac{2\eta_0}{\eta_1\gamma^2\alpha}\left(1 - \frac{\eta_1\alpha}{2}\right)\left(\mathbb{E}\|x^{t+1}_\phi - x^*_{\Phi,t}\|^2 + \mathbb{E}\|y_{t+1} - y^*_t\|^2\right)
$$
$$
+ \frac{2\eta_0}{\eta_1\gamma^2\alpha}\left(1 - \frac{\eta_1\alpha}{2}\right)\left(\mathbb{E}\|x^{t+1}_\psi - x^*_{\Psi,t}\|^2 + \mathbb{E}\|y_{t+1} - y^*_t\|^2\right) \tag{22}
$$
$$
+ \left(\frac{4\eta_0^3}{\eta_1^2\gamma^4\alpha(1/\gamma - \delta_\phi)^3} + \frac{4\eta_0^3}{\eta_1^2\gamma^4\alpha(1/\gamma - \delta_\psi)^3} + \frac{2L^2_{\phi,yx}\eta_0^3}{\eta_1^2\mu_\phi^3\gamma^4\alpha(1/\gamma - \delta_\phi)^2}\right.
$$
$$
\left. + \frac{2L^2_{\psi,zx}\eta_0^3}{\eta_1^2\mu_\psi^3\gamma^4\alpha(1/\gamma - \delta_\psi)^2} - \frac{\eta_0}{4}\right)\mathbb{E}\|G_{t+1}\|^2
$$
$$
- \frac{\eta_0}{2}\mathbb{E}\|\nabla F_\gamma(x_t)\|^2 + \frac{24\eta_0\eta_1 M^2}{\gamma^2\alpha} + \frac{24\eta_0\eta_1 M^2}{\gamma^2\alpha}.
$$

We now introduce a potential function

$$
P_t = \frac{2\eta_0}{\eta_1\gamma^2\alpha}\left(\mathbb{E}\|x^{t+1}_\phi - x^*_{\Phi,t}\|^2 + \mathbb{E}\|y_{t+1} - y^*_t\|^2 + \mathbb{E}\|x^{t+1}_\psi - x^*_{\Psi,t}\|^2 + \mathbb{E}\|z_{t+1} - z^*_t\|^2\right), \tag{23}
$$

and rewrite inequality (22) as

$$
\mathbb{E}[F_\gamma(x_{t+1})] + P_{t+1}
$$
$$
\leq \mathbb{E}[F_\gamma(x_t)] + (1 - \beta)P_t - \beta\mathbb{E}\|\nabla F_\gamma(x_t)\|^2 + \frac{48\eta_0\eta_1 M^2}{\gamma^2\alpha}
$$
$$
+ \left(\frac{\eta_0^3}{\eta_1^2\gamma^4\alpha^4} + \frac{L^2_{\phi,yx}\eta_0^3}{\eta_1^2\mu_\phi^3\gamma^4\alpha^3} + \frac{L^2_{\psi,zx}\eta_0^3}{\eta_1^2\mu_\psi^3\gamma^4\alpha^3} - \frac{\eta_0}{4}\right)\mathbb{E}\|G_{t+1}\|^2,
$$

where
$$\beta = \min\left\{\frac{\eta_1\alpha}{2}, \frac{\eta_0}{2}\right\}. \tag{24}$$

This inequality, together with the choice of $\eta_0$ and $\tau$ specified in this theorm, yields

$$E[F_\gamma(x_{t+1})] + P_{t+1} \leq \mathbb{E}[F_\gamma(x_t)] + (1-\beta)P_t - \beta\mathbb{E}\|\nabla F_\gamma(x_t)\|^2 + \frac{48\eta_0\eta_1 M^2}{\gamma^2\alpha}.$$

Taking average of these inequalities over $t = 0, \ldots, T-1$ yields

$$\frac{1}{T}\sum_{t=0}^{T-1}(P_t + \mathbb{E}\|\nabla F_\gamma(x_t)\|^2) \leq \frac{F_\gamma(x_0) - F_\gamma^* + P_0}{\beta T} + \frac{48\eta_0\eta_1 M^2}{\beta\gamma^2\alpha}, \tag{25}$$

where we use $F_\gamma^* \leq F_\gamma(x_T)$ due to Assumption 4.1(iii). Recall that $\eta_0 = \tau\eta_1$ and $\nu = \min\{1, \frac{2\tau}{\gamma^2\alpha}\}$. Using these, (23) and (25), we have

$$\frac{1}{T}\sum_{t=0}^{T-1}(\mathbb{E}\|x_\phi^{t+1} - x_{\Phi,t}^*\|^2 + \mathbb{E}\|x_\psi^{t+1} - x_{\Psi,t}^*\|^2 + \mathbb{E}\|\nabla F_\gamma(x_t)\|^2)$$

$$\leq \frac{1}{\nu T}\sum_{t=0}^{T-1}(P_t + \mathbb{E}\|\nabla F_\gamma(x_t)\|^2) \leq \frac{F_\gamma(x_0) - F_\gamma^* + P_0}{\nu\beta T} + \frac{48\eta_0\eta_1 M^2}{\nu\beta\gamma^2\alpha}.$$

By (24) and the choice of $\alpha$, $\eta_0$ and $\eta_1$ specified in this theorem, one has

$$\frac{\min\{1,\gamma^{-2}\}\nu\beta\gamma^2\alpha\epsilon^2}{384\eta_0 M^2} = \frac{\min\{1,\gamma^2\}\min\left\{\frac{\eta_1\alpha}{2}, \frac{\eta_1\tau}{2}\right\}\nu\alpha\epsilon^2}{384\eta_1\tau M^2} = \frac{\min\{1,\gamma^2\}\min\{\alpha,\tau\}\nu\alpha}{768\tau M^2}\epsilon^2 \geq \eta_1,$$

which implies that

$$\frac{48\eta_0\eta_1 M^2}{\nu\beta\gamma^2\alpha} \leq \min\{1,\gamma^{-2}\}\frac{\epsilon^2}{8}.$$

Suppose that $T$ satisfies (15). It then follows from (24), $\eta_0 = \tau\eta_1$, and the expression of $\eta_1$ that

$$T \geq \frac{16(F_\gamma(x_0) - F_\gamma^* + P_0)}{\min\{1,\gamma^{-2}\}\min\{\alpha,\tau\}\nu\epsilon^2}\max\left\{\frac{2}{\gamma^2(1/\gamma - \delta_\phi)}, \frac{2}{\gamma^2(1/\gamma - \delta_\psi)}, 2L_F\tau, \frac{768\tau M^2}{\min\{1,\gamma^2\}\min\{\alpha,\tau\}\nu\alpha\epsilon^2}\right\}$$

$$= \frac{8(F_\gamma(x_0) - F_\gamma^* + P_0)}{\min\{1,\gamma^{-2}\}\nu\beta\epsilon^2},$$

which implies that

$$\frac{F_\gamma(x_0) - F_\gamma^* + P_0}{\nu\beta T} \leq \min\{1,\gamma^{-2}\}\frac{\epsilon^2}{8}.$$

Hence, for any $T$ satisfying (15), one has

$$\frac{1}{T}\sum_{t=0}^{T-1}(\mathbb{E}\|x_\phi^{t+1} - x_{\Phi,t}^*\|^2 + \mathbb{E}\|x_\psi^{t+1} - x_{\Psi,t}^*\|^2 + \mathbb{E}\|\nabla F_\gamma(x_t)\|^2) \leq \min\{1,\gamma^{-2}\}\frac{\epsilon^2}{4},$$

which together with $x_{\Phi,t}^* = \text{prox}_{\gamma\Phi}(x_t)$ and $x_{\Psi,t}^* = \text{prox}_{\gamma\Psi}(x_t)$ yields

$$\frac{1}{T}\sum_{t=0}^{T-1}(\mathbb{E}\|x_\phi^{t+1} - \text{prox}_{\gamma\Phi}(x_t)\|^2 + \mathbb{E}\|x_\psi^{t+1} - \text{prox}_{\gamma\Psi}(x_t)\|^2 + \mathbb{E}\|\nabla F_\gamma(x_t)\|^2) \leq \min\{1,\gamma^{-2}\}\frac{\epsilon^2}{4}.$$

Since $\bar{t}$ is uniformly sampled from $\{1, \ldots, T\}$, we have

$$\mathbb{E}[\|x_\phi^{\bar{t}} - \text{prox}_{\gamma\Phi}(x_{\bar{t}-1})\|^2 + \|x_\psi^{\bar{t}} - \text{prox}_{\gamma\Psi}(x_{\bar{t}-1})\|^2 + \|\nabla F_\gamma(x_{\bar{t}-1})\|^2] \leq \min\{1,\gamma^{-2}\}\frac{\epsilon^2}{4}.$$

It then follows from Lemma 3.4 that $x_\phi^{\bar{t}}$ and $x_\psi^{\bar{t}}$ are both nearly $\epsilon$-critical points of problem (1).

$\square$

### A.3 Proof of Corollary 4.7

We present a detailed version of Corollary 4.7

**Corollary A.3.** *Consider Problem 2 and assume Assumption 4.6 holds. Suppose that the parameters $\gamma$, $\eta_0$ and $\eta_1$ in Algorithm 2 are chosen as follows:*

$$0 < \gamma < \min\{\delta_\phi^{-1}, \delta_\psi^{-1}\}, \quad \alpha = \min\left\{\frac{1/\gamma - \delta_\phi}{2}, \frac{1/\gamma - \delta_\psi}{2}\right\}, \quad \tau = \frac{\gamma^2\alpha^2}{4}, \quad \nu = \min\left\{1, \frac{2\tau}{\gamma^2\alpha}\right\},$$

$$\eta_1 = \min\left\{\frac{\gamma^2(1/\gamma - \delta_\phi)}{2}, \frac{\gamma^2(1/\gamma - \delta_\psi)}{2}, \frac{1}{2L_F\tau}, \frac{\min\{1, \gamma^2\}\min\{\alpha, \tau\}\nu\alpha}{768\tau M^2}\epsilon^2\right\}, \quad \eta_0 = \tau\eta_1.$$

*Then we have*

$$\frac{1}{T}\sum_{t=0}^{T-1}(\mathbb{E}\|x_\phi^{t+1} - prox_{\gamma\phi}(x_t)\|^2 + \mathbb{E}\|x_\psi^{t+1} - prox_{\gamma\psi}(x_t)\|^2 + \mathbb{E}\|\nabla F_\gamma(x_t)\|^2) \leq \min\{1, \gamma^{-2}\}\frac{\epsilon^2}{4},$$

*and consequently $x_\phi^{\bar{t}}$ and $x_\psi^{\bar{t}}$ are both nearly $\epsilon$-critical points of problem (2), whenever*

$$T \geq \frac{16(F_\gamma(x_0) - F_\gamma^* + P_0)}{\min\{1, \gamma^{-2}\}\min\{\alpha, \tau\}\nu\epsilon^2}\max\left\{\frac{2}{\gamma^2(1/\gamma - \delta_\phi)}, \frac{2}{\gamma^2(1/\gamma - \delta_\psi)}, 2L_F\tau, \frac{768\tau M^2}{\min\{1, \gamma^2\}\min\{\alpha, \tau\}\nu\alpha\epsilon^2}\right\}.$$

*with*

$$P_0 = \frac{2\tau}{\gamma^2\alpha}\left(\mathbb{E}\|x_\phi^1 - prox_{\gamma\phi}(x_0)\|^2 + \mathbb{E}\|x_\psi^1 - prox_{\gamma\psi}(x_0)\|^2\right).$$

Since problem (2) and Algorithm 2 are special cases of problem (1) and Algorithm 1 respectively, Corollary A.3 directly follows from Theorem A.2.

### A.4 Proof of Corollary 4.9

We present a detailed version of Corollary 4.9

**Corollary A.4.** *Consider Problem 3 and assume Assumption 4.8 holds. Suppose that the parameters $\gamma$, $\eta_0$ and $\eta_1$ in Algorithm 3 are chosen as follows:*

$$0 < \gamma < \delta_\phi^{-1}, \quad \alpha = \min\left\{\frac{1/\gamma - \delta_\phi}{2}, \mu_\phi\right\}, \quad \tau = \min\left\{\frac{\gamma^2\alpha^2}{4}, \frac{\mu_\phi^{1.5}\gamma^2\alpha^{1.5}}{4L_{\phi,yx}}\right\}, \quad \nu = \min\left\{1, \frac{2\tau}{\gamma^2\alpha}\right\},$$

$$\eta_1 = \min\left\{\frac{\gamma^2(1/\gamma - \delta_\phi)}{2}, \frac{1}{2L_F\tau}, \frac{\min\{1, \gamma^2\}\min\{\alpha, \tau\}\nu\alpha}{384\tau M^2}\epsilon^2\right\}, \quad \eta_0 = \tau\eta_1.$$

*Then we have*

$$\frac{1}{T}\sum_{t=0}^{T-1}(\mathbb{E}\|x_\phi^{t+1} - prox_{\gamma F}(x_t)\|^2 + \mathbb{E}\|\nabla F_\gamma(x_t)\|^2) \leq \min\{1, \gamma^{-2}\}\frac{\epsilon^2}{4},$$

*and consequently $x_{\bar{t}}$ is a nearly $\epsilon$-critical point of problem (3), whenever*

$$T \geq \frac{16(F_\gamma(x_0) - F_\gamma^* + P_0)}{\min\{1, \gamma^{-2}\}\min\{\alpha, \tau\}\nu\epsilon^2}\max\left\{\frac{2}{\gamma^2(1/\gamma - \delta_\phi)}, 2L_F\tau, \frac{384\tau M^2}{\min\{1, \gamma^2\}\min\{\alpha, \tau\}\nu\alpha\epsilon^2}\right\},$$

*with*

$$P_0 = \frac{2\eta_0}{\eta_1\gamma^2\alpha}\left(\mathbb{E}\|x_\phi^1 - prox_{\gamma F}(x_0)\|^2 + \mathbb{E}\|y_1 - y_0^*\|^2\right).$$

*Proof.* This proof is similar to that of Theorem A.2 except that the inequality (18) is replaced by

$$\|\nabla F_\gamma(x_t) - G_{t+1}\|^2 = \left\|\frac{1}{\gamma}(x_t - prox_{\gamma F}(x_t)) - \frac{1}{\gamma}(x_t - x_\phi^{t+1})\right\|^2$$

$$= \frac{1}{\gamma^2}\|prox_{\gamma F}(x_t) - x_\phi^{t+1}\|^2.$$

$\square$

# B  More Experimental Results

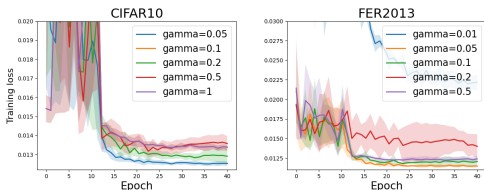

Figure 2: Ablation Study of SMAG for PU Learning

Table 4: Mean ± std of fairness results on CelebA test dataset with *Bags Under Eyes* task labels, and *Male* sensitive attribute. Results are reported on 3 independent runs. We use bold font to denote the best result and use underline to denote the second best.

| Methods | Bags Under Eyes, Male | | | |
|---|---|---|---|---|
| | pAUC↑ | EOD↓ | EOP↓ | DP ↓ |
| SOPA | 0.8293 ± 0.006 | 0.2015 ± 0.041 | 0.1000 ± 0.043 | 0.4055 ± 0.027 |
| SMAG* | 0.8261 ± 0.004 | 0.1848 ± 0.023 | 0.1065 ± 0.046 | 0.3754 ± 0.033 |
| SGDA | **0.8307** ± 0.003 | 0.2026 ± 0.028 | 0.1096 ± 0.031 | 0.4028 ± 0.039 |
| EGDA | 0.8262 ± 0.004 | 0.2223 ± 0.032 | 0.1287 ± 0.038 | 0.4200 ± 0.024 |
| SMAG | 0.8278 ± 0.002 | **0.1642** ± 0.025 | **0.0982** ± 0.034 | **0.3690** ± 0.029 |

