# OpenReview forum: "Single-Loop Stochastic Algorithms for Difference of Max-Structured Weakly Convex Functions"
_NeurIPS.cc/2024/Conference — NeurIPS 2024 poster_

### Official Review · Reviewer_gunK · 2024-06-30

**Soundness:** 3
**Presentation:** 3
**Contribution:** 3
**Rating:** 6
**Confidence:** 3

**Summary:**

This paper introduces a novel optimization problem termed Difference of Max-Structured Weakly Convex Functions (DMax). The DMax problem extends traditional frameworks such as difference-of-weakly-convex (DWC) optimization and weakly-convex-strongly-concave (WCSC) min-max optimization, which are widely utilized in the machine learning community.

Using a Moreau envelope smoothing technique, the authors propose a stochastic algorithm called SMAG for optimizing DMax in non-smooth settings. This approach not only addresses the DMax problem but also applies to DWC and non-smooth WCSC min-max optimization scenarios, demonstrating comparable convergence rates.

Experimental results validate the efficacy of their algorithms across various applications, including Positive-Unlabeled (PU) Learning and partial Area Under the Curve (AUC) optimization with an adversarial fairness regularizer.

**Strengths:**

**Originality:**
The paper introduces a new optimization problem termed Difference of Max-Structured Weakly Convex Functions (DMax), which extends existing frameworks like difference-of-weakly-convex (DWC) optimization and weakly-convex-strongly-concave (WCSC) min-max optimization. This extension demonstrates a significant level of originality in problem formulation within the field of optimization. By addressing a broader class of problems under the DMax framework, the paper contributes novel insights and methodologies that enrich the theoretical foundations of optimization in machine learning and related fields.

**Quality:**
The paper maintains high-quality standards across various aspects. It rigorously establishes the theoretical foundations of the DMax problem and proposes a Moreau envelope smoothing technique for optimization in non-smooth settings. The SMAG algorithm introduced is methodically developed and supported by well-defined lemmas and references, ensuring robustness and reliability in the proposed approach, though I don't check the proofs in detail.
Furthermore, the experimental validation on applications such as Positive-Unlabeled Learning and partial AUC optimization with adversarial fairness regularization underscores the practical relevance and effectiveness of the proposed methods.

**Clarity:**
The clarity of the paper is a notable strength. It effectively communicates complex concepts and methodologies in a clear and accessible manner, making it understandable even for readers with limited expertise in the specific problem domain. The intuitive presentation of algorithms and key steps, coupled with clear explanations of referenced lemmas, enhances readability without sacrificing depth. This clarity not only facilitates comprehension but also promotes transparency in the theoretical derivations and experimental procedures, thereby bolstering the paper's overall impact.

**Significance:**
By addressing challenging optimization problems and providing practical algorithms validated in real-world scenarios, the paper significantly advances both theoretical understanding and practical applications in machine learning optimization.

**Weaknesses:**

**Problem Statement and Novelty:**
The paper introduces a new problem that extends traditional frameworks such as difference-of-weakly-convex (DWC) optimization and weakly-convex-strongly-concave (WCSC) min-max optimization, presenting the SMAG algorithm as a solution. However, the paper could benefit from clearer justification of how this generalization represents a significant departure from incremental advancements. This clarity is pivotal for establishing the theoretical contribution within a novel setting.

**Complexity Analysis:**
Upon examining the complexity results, it is apparent that in non-convex, non-smooth min-max problems, SMAG demonstrates complexity similar to Epoch-GDA under comparable assumptions (Table 2). This finding raises significant concerns about the broader implications for WCSC min-max optimization, particularly considering the emphasis on complexity as a contribution (Line 59). A clearer explanation is needed on how SMAG surpasses existing complexities in this domain, beyond the practical advantage of a single-loop method for tuning, to strengthen the paper's impact solely from a complexity perspective.

**Algorithmic Technique:**
The proof and algorithmic technique employed in this study resemble SBCD, utilizing the Moreau envelope to transform non-smooth problems into smooth ones, thereby enhancing accessibility. Hence, it is crucial to clearly delineate why SMAG achieves superior complexity results compared to SBCD or to identify any overlooked aspects that may explain this discrepancy.

I welcome further discussion on these points to ensure a comprehensive evaluation. Thanks!

**Questions:**

In the experiments section, the authors employ a linear classification model with hinge loss (Line 271). However, this setting appears to be a smooth one, potentially limiting the demonstration of the benefits of the authors' method in non-smooth DWC problems, despite SMAG showing superiority over other methods.
Regarding the large variance in training curves observed, such as in FER 2013 (Figure 1), it raises questions about the stability and robustness of the method in practical applications. To enhance confidence in the results, I suggest exploring parameter tuning to achieve more stable and convincing outcomes.

**Limitations:**

The paper is theoretical, and I believe there is no need to confirm societal impact.

---

> ### Author Rebuttal · Authors · 2024-08-06
>
> We thank the reviewer for the positive comments and feedback on our paper.
>
> **Q1:** Clarification of  theoretical contribution within a novel setting.
>
> **A.** We proposed a unified framework of analysis for non-smooth DWC and WCSC min-max problems, which leads to single-loop methods that achieves the best convergence rate. Our main contribution is the novel analysis that applies to problems with complex structures (difference of max-structured weakly convex functions) and potential non-smoothness.  In particular, in our technical analysis, the Lemma 4.4 is novel, which does not exist in any existing work. This make its possible for us to prove the convergence of a special error function $\|x_\phi^t - prox_{\gamma\phi}(x_{t-1})\|^2 + \|x_\psi^t - prox_{\gamma\psi}(x_{t-1})\|^2 + \|\nabla F_{\gamma}(x_t)\|^2$.
>
> **Q2:** Confusion about the complexity as a contribution (Line 59) for WCSC problems.
>
> **A:** We are sorry for the confusion. For WCSC problem, we did not intend to claim a better complexity, but rather emphasize that it is the first single-loop method with the same complexity as existing double-loop methods. The improved complexity of our algorithm lies at its application to difference-of-weakly convex functions, which can be seen from Table 1.
>
>
> **Q3:** Does the proof and algorithmic technique employed in this study resemble SBCD?
>
> **Response.** NO. The main difference between the analysis of SMAG and SBCD is in the error bounds $E\|x_\phi^{t+1} - prox_{\gamma \Phi}(x_t)\|^2$ and $E\|x_\psi^{t+1} - prox_{\gamma \Psi}(x_t)\|^2$. For SBCD, they use inner loops to solve for $prox_{\gamma \Phi}(x_t)$ and $prox_{\gamma \Psi}(x_t)$. In order to reach error bounds $E\|x_\phi^{t+1} - prox_{\gamma \Phi}(x_t)\|^2 \leq O(\frac{1}{t+1})$ and $E\|x_\psi^{t+1} - prox_{\gamma \Psi}(x_t)\|^2 \leq O(\frac{1}{t+1})$ at iteration $t$, they need to use $O(t^2)$ inner loop iterations because of the involved maximization. With $O(\epsilon^{-2})$ outer iterations guarantees a nearly $\epsilon$-critical point, it leads to a total sample complexity of $O(\epsilon^{-6})$. In contrast, our analysis directly builds the recursion for $\|x_\phi^{t+1} - prox_{\gamma \Phi}(x_t)\|^2 + \|y_{t+1} - y^*(prox_{\gamma\phi}(x_{t}))\|^2$ in Lemma 4.4. In addition, we introduce a special potential function $P_t$ as in Eq. (23). We are able to build a decreasing recursion in terms of $P_t$, i.e., $P_{t+1}$ in the left, and $(1-\beta)P_t$ in the right of the bound (cf line 541), where $\beta<1$. These make it possible for us to prove a better  convergence rate of a special error function $E[\|x_\phi^t - prox_{\gamma\phi}(x_{t-1})\|^2 + \|x_\psi^t - prox_{\gamma\psi}(x_{t-1})\|^2 + \|\nabla F_{\gamma}(x_t)\|^2]$ for a randomly chosen $t$.
>
>
> **Q4.** Misunderstanding about the hinge loss (Line 271) as a smooth function. About the large variance in training curves observed, such as in FER 2013 (Figure 1).
>
> **A.** The hinge loss is in the form of $\ell(x,y) = \max(0, 1- y * f(x))$ where $x,y$ are the data sample and its label, and $f(x)$ is the prediction score of the sample $x$.  This clearly is a non-smooth loss due to the presence of the function $\max(0, \cdot)$. Regarding the variance in the training curves: (i) the large variance of our method for FER2013 is due to a bug in the code, which loads a wrong file of one trial result for plotting. A corrected training curve of our method is included in the attached PDF file of the global rebuttal. It has much smaller variances. (ii) We have also incldued the means and standard deviations of our method on all datasets for references in the attached PDF file. It can be seen that the standard deviations are indeed one order smaller than the loss values. Please also note that the scale of the y-axis is very small, which makes the std more obvious. (iii) We have conducted hyperparameter tuning as stated in lines 283-289. Thank you!

---

> > ### Comment · Reviewer_gunK · 2024-08-10
> >
> > Thank you for the clear response. I have slightly increased my score, particularly for the clarification of the main difference between the analysis of SMAG and SBCD, which serves as the new proof technique.

---

> > > ### Author Response · Authors · 2024-08-12
> > >
> > > Thank you for acknowledging our responses. We are glad to hear that it addresses your concerns.

---

### Official Review · Reviewer_QoFj · 2024-07-14

**Soundness:** 3
**Presentation:** 3
**Contribution:** 3
**Rating:** 5
**Confidence:** 1

**Summary:**

This paper proposes a stochastic Moreau envelope approximate gradient method dubbed SMAG, the first single-loop algorithm for solving these problems, and provides a state-of-the-art non-asymptotic convergence rate.
This paper achieves the best complexity of order $O(\epsilon^{-4})$.
Futhermore, the agorithm of this paper only use a single loop which is easy to implement and tune the hyperparameters.

A typo: Line 3, it should be $\Psi(x) = \max_z \psi(x, z)$.

**Strengths:**

No

**Weaknesses:**

No

**Questions:**

This paper considers the problem of a special structure that is the difference of convex functions. However, in this paper, this special structure seems not be used.
As I know, for the problem of  difference of convex functions, there are several algorithms can achieve the faster convergenc rate of general non-convex functions.
So, I wonder whether the results of this paper is useful enough.

---

> ### Author Rebuttal · Authors · 2024-08-06
>
> We thank the reviewer for the positive comments and feedback on our paper.
>
> **Q:** The special structure of the difference of convex functions is not used.
>
> **A:** Thank you for your good question. (1) In this paper the problem considered is the Difference of Max-Structured Weakly Convex Functions (DMax) Optimization, which is more general than the difference of convex functions due to the maximization structure in the two component functions. Hence, we cannot use some technique like linearization of the second component as in previous work on solving difference-of-convex functions.  (2) We do utilize the difference structure to some degree. In particular, we apply Moreau envelope smoothing to each component separately instead of jointly. It is notable that even each component is weakly convex, their difference is not necessarily weakly convex. Hence, applying Moreau envelope smoothing to the joint function would not make sense.  (3) For the difference of weakly-convex functions problem (DWC), we have summarized its existing works in Table 1. To the best of our knowledge, our proposed method is the first single-loop stochastic methods and it achieves the best convergence rate $O(\epsilon^{-4})$.

---

### Official Review · Reviewer_2iRa · 2024-07-20

**Soundness:** 3
**Presentation:** 3
**Contribution:** 3
**Rating:** 6
**Confidence:** 3

**Summary:**

The paper considers the problem of minimizing a difference of maximum of two functions, under various regularity assumptions. Two important settings are difference of weakly convex functions and weakly convex strongly concave min-max problems. The authors propose a single loop algorithm with a convergence rate (defined in an appropriate sense) of $\epsilon^{-4}$. The authors apply their method to positive-unlabeled (PU) learning and partial area under ROC curve (pAUC) optimization with an adversarial fairness regularizer, and compare against existing methods.

**Strengths:**

1. The paper is generally well written. The authors motivate the technical challenges well and how to get around them rather.
2. Getting single loop algorithms with the Moreau smoothing technique is challenging. The authors make important contributions in this regard.
3. The experimental results are encouraging. The authors show consistent improvement in their applications over the baselines.

**Weaknesses:**

1. (Writing) The authors should distinguish if the difference/improvement from previous work are "double loop vs single loop", and/or faster rates and/or weaker assumptions. For the DWC setting, it seems it is "double loop vs single loop", and/or faster rates and weaker assumptions, where as for WCSC, it is "double loop vs single loop".

2. (Writing) The authors should mention the notion of optimalities in various settings ($\epsilon$ nearly critical and $\epsilon$ near stationarity) eraly on, and remark that they are standard.

**Questions:**

1. Are the proof ideas useful to give single loop algorithms for non-smooth convex minimization via Moreau smoothing, with optimal rates?

2. Optimality: Is the $\epsilon^{-4}$ rate achieved optimal in these settings?

---

> ### Author Rebuttal · Authors · 2024-08-06
>
> We thank the reviewer for the positive comments and feedback on our paper.
>
> **Q1.** The authors should distinguish if the difference/improvement from previous work are "double loop vs single loop", and/or faster rates and/or weaker assumptions. For the DWC setting, it seems it is "double loop vs single loop", and/or faster rates and weaker assumptions, where as for WCSC, it is "double loop vs single loop".
>
> **A:** The improvement over baselines can be seen from the difference in terms of assumptions, complexity and number of loops in Table 1 and Table 2. For the DWC setting, our improvement over SDCA [25] lies at weaker assumption and single loop, over SSDC-X lies at weaker assumption, faster rate and single loop, over SBCD lies at faster rate and single loop. In the WCSC setting, our improvement over PG-SMD/Epoch-GDA lies at single loop, over SAPD+ lies at weaker assumption and single loop, and over StocAGDA lies at weaker assumption.
>
>
> **Q2.** The authors should mention the notion of optimalities in various settings (nearly critical and  near stationarity) eraly on, and remark that they are standard.
>
> **A:** Thank you the suggestion. We will revise our draft accordingly.
>
>
> **Q3.** Are the proof ideas useful to give single loop algorithms for non-smooth convex minimization via Moreau smoothing, with optimal rates?
>
> **A:** Thanks for the good question! We have not explored the non-smooth convex minimization via Moreau smoothing. It might be useful in the sense that our technique can be used to argue that with one step update of $x_t'$ for solving  $\min_{x'}f(x') + \frac{1}{2\gamma}\|x_t - x'\|^2$, we can establish recursion of $|x'_t -  $
>
> $ prox_{\gamma f}(x_t)\|^2$, where $prox_{\gamma f}(x_t)$ is the optimal solution to  $\min_{x}f(x') + \frac{1}{2\gamma}\|x_t - x'\|^2$. This is the error of gradient estimator of the function $F_{\gamma}(x_t) = \min_{x}f(x') + \frac{1}{2\gamma}\|x_t - x'\|^2$. For deep analysis, we will leave it as a future work.
>
>
> **Q4.** Is the $O(\epsilon^{-4})$ rate achieved optimal in these settings?
>
> **Response.** We believe so. $O(\epsilon^{-4})$ rate has been proved to the optimal even in the smooth setting for non-convex optimization [r1].
>
>
> Reference.
>
>
> [r1]:  Arjevani et al. Lower bounds for non-convex stochastic optimization.  2019.

---

### Official Review · Reviewer_jGsv · 2024-07-30

**Soundness:** 3
**Presentation:** 3
**Contribution:** 3
**Rating:** 5
**Confidence:** 2

**Summary:**

The paper studies a class of optimization problem named DMax, i.e., to minimize a loss that is a difference of two max functions. This problem covers both difference of weakly-convex optimization and weakly-convex strongly-concave minimax optimization. Existing algorithms require double-loop structure to solve the inner problem within certain accuracy. This paper proposes a simple single-loop stochastic algorithm that achieves state-of-the-art convergence rates. Empirical experiments demonstrate the effectiveness of the proposed algorithm.

**Strengths:**

The paper considers a class of difference of max-structured weakly-convex optimization problems, which can be interesting and novel in its problem fomulation. The proposed algorithm is simple to implement as it is single-loop structured and also clear to understand. The convergence rate matches exsiting works that use more complicated structures of the algorithms.

**Weaknesses:**

1. There are some typos that may affect readibility. (1) Is the definition of $\Psi$ in line 3 of the abstract correct? Shouldn't it be $\max_z\psi$? (2) I guess that it should be "Single-Loop" in the title? (3) In line 27, there is an empty citation for adversarial learning. (4) In line 49, "sufficient to to". (5) "covnex" in line 4 of Table 2. (6) Should it be $\partial_x \psi$ in line 4 of Algorithm 1 and line 3 of Algorithm 2?

2. Although the problem fomulation is interesting in optimization theory, the current applications seem to be limited to only applications of the special cases DWC and WCSC min-max optimization. Therefore, the problem can be a little bit artificial in the sense that these applications do not necessarily require such a formulation. It can be done by studying DWC and min-max separately, and the major benefit is that DMax provides a unified way. Are there examples that DMax gives more applications beyond DWC and min-max optimization?

3. (Minor) Although I like the proposed algorithm, which is simple and clear to understand, it seems like all the technical steps exist before. The paper combines these known results and uses it for a new class of problem. For example, it is well-known that single-loop updates can give the same rate for strongly-concave problems compared to double-loop algorithms that first solve the inner-problems to a required accuracy. It is also known that weakly-convex problems can be handled using Moreau envelope and usually enjoy the same rate as smooth problems. As a result, the convergence rates and theoretical analysis in the paper are not surprising.

I am the emergence reviewer. I do not carefully check all the technical details and proofs in the paper. However, the algorithmic ideas and analysis outline look good to me. I will be on the positive side with a low confidence rate.

**Questions:**

See weaknesses.

**Limitations:**

See weaknesses.

---

> ### Author Rebuttal · Authors · 2024-08-06
>
> We thank the reviewer for the positive comments and feedback on our paper.
>
> **Q1.** There are some typos that may affect readability
>
> **A:** Thank you for pointing out these typos. We will fix them in our revision.
>
>
> **Q2.** Are there examples that DMax gives more applications beyond DWC and min-max optimization?
>
> **A:** One example can be found in [43] for optimizing partial AUC in a range of false positive rates for binary classification. The objective can be written as
> $\sum_{i=1}^{N^+}\phi_{n}(s_{i}) - \phi_m(s_{i})$, where $s_i=(s_{i1},\ldots, s_{iN^-})$ denotes the pairwise loss between a positive data $x_i$ and all $N^-$ negative samples, and $\phi_n$ is the an operator that outputs the sum of the top-$n$ items in the input vector. Since $\phi_{n}(s_{i}) =\max_{p_j\geq 0, \sum p_j = n}p_j s_{ij}$, as a result, $\phi_{n}(s_{i}) - \phi_m(s_{i})$ is a structure of DMax.
>
>
> **Q3:** About the technical novelty.
>
> **A:** It is true that single-loop updates can give the same rate for strongly-concave problems compared to double-loop algorithms that first solve the inner-problems to a required accuracy. However, existing single-loop algorithms for min-max problems all require smoothness of the objective function. In our work, we did not use the smoothness assumption in terms of $x$ and hence the analysis is completely different. In terms of technical analysis, the Lemma 4.4 is novel, which does not exist in any existing work. In addition, we introduce a special potential function $P_t$ as in Eq. (23).  These make it possible for us to prove the convergence of a special error function $E[\|x_\phi^t - prox_{\gamma\phi}(x_{t-1})\|^2 + \|x_\psi^t - prox_{\gamma\psi}(x_{t-1})\|^2+ \|\nabla F_{\gamma}(x_t)\|^2]\leq \epsilon$ for a randomly chosen $t$.

---

> > ### Comment · Reviewer_jGsv · 2024-08-12
> >
> > Thanks for your response! I have no additional problems. I will keep my score, which is on the positive side. I am willing to support the acceptance of the paper.

---

> > > ### Author Response · Authors · 2024-08-12
> > >
> > > Thank you for acknowledging our responses and the positive feedback.

---

### Author Rebuttal · Authors · 2024-08-06

We thank the reviewers for their valuable feedback. As Reviewer gunK noted, the training curve of SMAG for the FER2013 dataset in Figure 1 exhibits unusually high variance. After careful checking, we identified a bug in the code, which loads a wrong file of one trial result for plotting. We have included the corrected figure in the attached pdf file. We also include the mean and std of loss values of our method on different datasets in the file.

---

### Decision · Program_Chairs · 2024-09-25

**Decision:**

Accept (poster)

**Comment:**

Paper proposes a new Moreau envelope based algorithm for solving a new class of optimization problems called Difference of Max-structured weakly-convex functions (DMax). This generalizes two others problems: Difference of Weakly-Convex functions (DWC) and Weakly-Convex-Strongle-Concave (WCSC) problems. The proposed algorithm is a single loop algorithm, which is more practical than multi-loop methods. It is proven to achieve new state-of-the-art gradient complexity for DWC problems and known state-of-the-art results for WCSC problems. Central idea is to optimize a proxy objective which is difference of Moreau envelopes than optimize the Moreau envelope of the difference. Paper also provides experimental comparison of their algorithm on two different ML problem.

Obtaining single loop algorithm for Moreau smoothing is known to be challenging. Paper is mostly well written and experimental results are promising. Reviewers also raised a concern that most of the given applications are special cases which are not as general as the formulated problems. This was not satisfactorily addressed. There were also few typos affecting the readability. However, most reviewers recommended acceptance of this paper.